



# Wind tunnel study on power and loads optimization of two yaw-controlled model wind turbines

Jan Bartl[1], Franz Mühle[2], and Lars Sætran[1]

[1] Department of Energy and Process Engineering, Norwegian University of Science And Technology, Trondheim, Norway
[2] Faculty of Environmental Sciences and Natural Resource Management, Norwegian University of Life Sciences, Ås, Norway

*Correspondence to:* Jan Bartl (jan.bartl@ntnu.no)

**Abstract.** In this experimental wind tunnel study the effects of intentional yaw misalignment on the power production and loads of a downstream turbine are investigated for full and partial wake overlap situations. Power, thrust force and yaw moment are measured on both the upstream and downstream turbine. The influence of inflow turbulence level and streamwise turbine separation distance are analyzed for full wake overlap situations. For partial wake overlap the concept of downstream turbine yawing for yaw moment mitigation is examined for different lateral offset positions.

Results indicate that upstream turbine yaw misalignment is able to increase the combined power production of the two turbines for both partial and full wake overlap setups. For aligned turbine setups the combined power is increased between 3.5% and 11% depending on the inflow turbulence level and turbine separation distance. The increase in combined power is at the expense of increased yaw moments on both upstream and downstream turbine. For partial wake overlap situations, yaw moments on the downstream turbine can be mitigated through upstream turbine yawing, while simultaneously increasing the combined power production. A final test case demonstrates the concept of opposed downstream turbine yawing in partial wake situations, which is shown to reduce its yaw moments and increasing its power production by up to 5%.

## 1 Introduction

In wind farms the individual wind turbines interact aerodynamically through their wakes. Besides significant power losses, rotors exposed to upstream turbines' wakes experience higher unsteady loading (Kim et al., 2015). The reduced power and increased rotor loads are dependent on the downstream turbine's lateral and streamwise location in the wake, the upstream turbine's control settings and the characteristics of the incoming wind. The inflow characteristics are governed by the atmospheric stability, in which the turbulence level as well as the degree of shear and veer are important parameters. In combination with the wind farm layout, the site dependent wind statistic, such as wind speed and direction distributions, define the occurrence for downstream turbines to be fully or partially exposed to the upstream turbine's wake.

In order to mitigate power losses and wake induced loads on downstream turbines, different upstream turbine control strategies have recently been suggested (Knudsen et al., 2014; Gebraad et al., 2015). These include methods to reduce the axial-induction of an upstream turbine and thus also mean and turbulent gradients in the wake (Annoni et al., 2016; Bartl and Sætran, 2016) as



well as wake redirection techniques (Fleming et al., 2015). The most discussed wake deflection mechanisms include individual pitch angle control, tilt angle variation and yaw angle actuation. In a computational fluid dynamics (CFD) study Fleming et al. (2015) compare these techniques with regards to power gains and blade out-of-plane bending loads on a two turbine setup. Individual pitch control was observed to cause high structural loads. Most current turbine designs do not feature tilt mechanisms,

while yaw actuation is concluded to be a promising technique due to its simple implementability. As all modern wind turbines are equipped with yaw actuators, intentional yaw misalignment can be used to laterally deflect the wake flow and potentially increase the wind farm power output.

A number of recent research focused on the wake characteristics behind a yawed wind turbine. In a combined experimental and computational study Howland et al. (2016) measured the wake of yawed small drag disc and conducted a Large-Eddy-

Simulation (LES) behind an actuator disc/line modeled rotor. They discussed different quantifications for wake deflection and characterized the formation of a curled wake shape due to a counter-rotating vortex pair. A similar wake shape was found in a LES study by Vollmer et al. (2016), who found a significant variation of wake shape and deflection depending on the atmospheric stability. The yawed wake characteristics' dependency on inflow turbulence and shear were investigated in an experimental study by Bartl et al. (2018). The inflow turbulence level was observed to influence the shape and deflection of

the wake, in contrast to a moderate shear in the inflow. Schottler et al. (2018) highlight the importance of considering non-Gaussian distributions of velocity increments in wind farm control and layout optimizations. A ring of strongly intermittent flow is shown to surround the mean velocity deficit locations, suggesting a much wider wake expansion as based on the mean velocity. An extensive theoretical and experimental study on yaw wakes was performed by Bastankhah and Porté-Agel (2016). They presented a theoretical description for the formation of the counter-rotating vortex pair in the wake and developed a so-

phisticated analytical model for the far wake of a yawed turbine. Including inflow turbulence as an additional input parameter makes Bastankhah and Porté-Agel's model a favorable alternative to the wake deflection model by Jiménez et al. (2010).

Moreover, various research investigated the potential of overall wind farm power gains through intentional yaw misalignment. An experimental study by Adaramola and Krogstad (2011) on two aligned model wind turbines ($x/D = 3$) demonstrated an increase in combined efficiency with increasing upstream turbine yaw angle. For a yaw angle of $30°$, they measured an in-

crease of 12% in combined power compared to the reference case at $0°$. For the same separation distance Schottler et al. (2015) measured a combined power increase of about 4% for an upstream turbine yaw angle of $-18°$. Their experimental study on two aligned model turbines furthermore pointed out clear asymmetries of the downstream turbine power output with regards to the upstream turbine yaw angle. Another experimental study on three model wind turbines was presented by Campagnolo et al. (2016), who measured a combined power increase of 21% for an lateral offset of $\Delta z/D = 0.45$ between the turbines.

Comprehensive studies on yaw misalignment for optimized full wind farm control haven been presented by Fleming et al. (2014) and Gebraad et al. (2016). They analyzed wake mitigation strategies by using both the LES code SOWFA as well as a parametric wake model. A dedicated full-scale study by McKay et al. (2013) investigated the connection of yaw alignment and power output of a downstream turbine operated in the wake of an upstream turbine. They found an independent yaw alignment for the purpose of individual power increase of downstream turbines operated in partial wake situations.

Most of these studies focus on the possibilities for power optimization through yaw control; however, the discussion of in-



creased structural loads is often left open. Yet, yaw misalignment of an undisturbed turbine was observed to create increased unsteady loading on the yawed rotor. In a simulation by Kragh and Hansen (2014) these loads are quantified for different inflow conditions. It is furthermore shown that load variations due to wind shear can potentially be alleviated by yaw misalignment. Load characteristics on a yawed model turbine rotor were compared to various computational approaches by Schepers

et al. (2014). The so-called Mexnext project revealed modeling deficiencies while shedding light on complex unsteady flow phenomena during yaw. In a recent paper by Damiani et al. (2017) damage equivalent loads and extreme loads under yaw misalignment are measured and predicted for a fully instrumented wind turbine. They observed rather complex, inflow-dependent load distributions for yaw angle offsets. In a computational setup of ten aligned, non-yawed wind turbines Andersen et al. (2017) recently investigated the influence of inflow velocity, turbulence intensity and streamwise turbine spacing on the yaw

moments and other equivalent loads on downstream turbines operated in the wake. The study shows up unexpected load peaks for every second or third downstream turbine in below-rated operating conditions. A way to utilize measured rotor loads such as yaw moments to estimate rotor yaw misalignment, inflow shear or partial wake rotor operation is investigated by Schreiber et al. (2016). Using a computational framework of a wake model, BEM model for power and loads and a gradient-based optimizer van Dijk et al. (2017) investigated the effects of yaw misalignment on power production and loads in full and partial

wake overlap situations. They found that upstream turbine yaw-misalignment is able to increase the total power production of their modeled wind farm, while reducing the loads in partial wake overlap situations.

The objective of the present study is to analyze potentials of yaw control for the often contradicting goals of combined power gains and load mitigation. Balancing the benefits of power gains and costs of increased rotor loads is of utmost importance for the design of cost-effective wind farm control strategies. For this purpose the parameters turbine separation distance $x/D$,

lateral turbine offset $\Delta z/D$ and turbine yaw settings $\gamma_{T1}$ and $\gamma_{T2}$ are systematically varied in this wind tunnel experiment. Special focus is given to the concept of downstream turbine yawing in partial wake situations for the purpose of load reduction and combined power gains. Together with the inflow-dependent wake flow measurements on the same experimental setup presented in Bartl et al. (2018), this study completes the link between detailed wake flow characteristics and power, yaw moments and thrust forces on a turbine operated in the wake.

## 2   Experimental setup

### 2.1   Wind turbine models

Two wind turbine models of the exactly same rotor geometry were used for this study. The rotor was designed based on the NREL S826 aifoil and has a total diameter of $D = 0.894\,m$. The tower and nacelle structure of the upstream turbine (T1) is slightly slimmer than that of the downstream turbine (T2), in order to minimize the effect on the wake flow behind the yawed

upstream turbine. The maximum power point of both turbines is reached at a tip speed ratio of $\lambda_{T1} = \lambda_{T2} = 6.0$ in undisturbed inflow. In this experiment T2 is controlled to its optimum power point, which strongly varies for different positions and upstream turbine operational parameters. The exact geometry and detailed performance curves of T1 are described in Bartl et al. (2018), while T2's characteristics can be found in Bartl and Sætran (2017). In contrast to most other turbines, the investigated





model turbines rotate counter-clockwise. Positive yaw is defined as indicated in Figure 2.

The experiments were performed in the closed-loop wind tunnel at the Norwegian University of Science and Technology (NTNU) in Trondheim, Norway. The tunnel's cross-section measures $2.71\,m$ in width, $1.81\,m$ in height and $11.15\,m$ in length. The turbine models are operated at a blade tip Reynolds numbers of approximately $Re_{\mathrm{tip}} \approx 10^5$.

Moreover, about $12.8\%$ of the wind tunnel's cross sectional area are blocked by the turbines' rotor swept area. The wind tunnel width measures about three times the turbine's rotor diameter, which leaves sufficient space for lateral wake deflection and offset positions for T2. However, a speed-up of the flow in free-stream areas around the rotors is observed due to blockage effects as described in detail in Bartl et al. (2018).

## 2.2 Inflow conditions

The influence of different inflow turbulence levels is investigated in this study. For this purpose the turbines are exposed to an inflow of very low turbulence intensity $TI_A = 0.23\%$ (Inflow A) as well as high turbulence intensity $TI_B = 10.0\%$ (Inflow B). Inflow B is generated by a static grid at the wind tunnel inlet. The grid-generated turbulence decays with increasing downstream distance to about $TI_B = 5.5\%$ at $x/D = 3$ and to $TI_B = 4.0\%$ at $x/D = 6$. The profiles of streamwise mean

velocity and turbulence intensity measured in the empty wind tunnel for different downstream positions are presented in Bartl et al. (2018). Inflow A is assessed to be uniform within $\pm 0.8\%$ over the rotor swept area. A velocity variation of $\pm 2.5\%$ is measured at $x/D = 0$ for Inflow B, as the footprint of the grid's single bars are still detectable At $x/D = 3$, however, the grid-generated turbulent flow is seen to be uniform within $\pm 1.0\%$. Both test cases were performed at the constant reference velocity of $u_{ref} = 10.0\,m/s$.

## 2.3 Measurement techniques

The mechanical power on both rotors was measured in separate steps with a HBM torque transducer of the type T20W-N/2-Nm, which is installed in the nacelle of the downstream turbine T2. The transducer is connected to the rotor shaft through flexible couplings. An optical photo cell inside the nacelle makes the rotor's rotational speed assessable. On the test rig of T1

the rotational speed is controlled via a servo motor, ensuring the same power and load characteristics as for T2.

For the purpose of thrust force and yaw moment measurements the model turbines are separately installed on a six-component force balance by Carl Schenck AG. By constantly recording signals obtained from the three horizontal force cells, the yaw moments referred to the rotor center can be calculated. For the assessment of the rotor thrust, the drag force on tower and nacelle is measured isolated and then subtracted from the total thrust. No such correction is applied for the assessment of the

yaw moments.





**Table 1.** Overview of test cases.

| Test case | | Parameter variation | Inflow turbulence | Yaw angle $\gamma_{T1}$ | Streamwise separation $x/D$ | Lateral offset $\Delta z/D$ | Yaw angle $\gamma_{T2}$ |
|---|---|---|---|---|---|---|---|
| 1 (a) | Aligned turbines | $\gamma_{T1}$ & $x/D$ | 0.23% | [-40°,..., +40°] | 3 & 6 | 0 | 0° |
| 1 (b) | Aligned turbines | $\gamma_{T1}$ & $x/D$ | 10.0% | [-40°,..., +40°] | 3 & 6 | 0 | 0° |
| 2 (a) | Offset turbines | $\Delta z/D$ | 10.0% | 0° | 3 | [-0.5,...+0.5] | 0° |
| 2 (b) | Offset turbines | $\Delta z/D$ | 10.0% | +30° | 3 | [-0.5,...+0.5] | 0° |
| 3 (a) | Downstream turbine yaw | $\Delta z/D$ & $\gamma_{T2}$ | 10.0% | 0° | 3 | [-0.5,...+0.5] | [-30°,...,+30°] |
| 3 (b) | Downstream turbine yaw | $\Delta z/D$ & $\gamma_{T2}$ | 10.0% | +30° | 3 | [-0.5,...+0.5] | [-30°,...,+30°] |

## 2.4 Statistical measurement uncertainties

The statistical measurement uncertainties for power coefficients, thrust coefficient and normalized yaw moments have been calculated following the procedure described by Wheeler and Ganji (2004). Random errors are computed from repeated measurements of various representative measurement points based on a 95 % confidence interval. Furthermore, the match of power and thrust values of the baseline cases (e.g. $\gamma_{T1} = 0°$, $x/D = 3$, $\Delta z/D = 0$) with previous results e.g. by Bartl and Sætran (2016, 2017) has been checked for consistency.

For the purpose of clarity, errorbars are not shown in the resulting graphs in Section 3. Instead, a short overview of uncertainties for the different measures is given here. The total uncertainty in T1's power coefficient is 0.011 (1.9%) for non-yawed operation, rising up to about 0.017 (3.9%) for a yaw angle of $\gamma_{T1} = 30°$. The uncertainty in T1's thrust coefficient is assessed to be very similar, varying from 0.013 (1.4%) to 0.018 (3.1%) for yaw angles 0° and ±40°, respectively. The uncertainty in normalized yaw moments $M_y^*$ is 0.0032, which corresponds to almost 15% of the absolute measurement value at $\gamma_{T1} = 30°$. Due to very small absolute values of the yaw moments, the relative uncertainty is rather high. In the case of T2, the uncertainties are presented representatively for the aligned test case, in which the upstream turbine is operated at $\gamma_{T1} = 30°$ and T2 located at $x/D = 3$ and operated at $\gamma_{T2} = 0°$. The total uncertainties in power and thrust coefficient are 0.006 (2.5% of the absolute $C_P$-value) respectively 0.007 (0.9% of the absolute $C_T$-value). The normalized yaw moment of the downstream turbine for this case is amounts 0.0019 (about 8% of the absolute value).

## 2.5 Test case definition

Three main test cases are investigated in this study. In a first test case the two model turbines are installed in an aligned arrangement in the wind tunnel, i.e. T2 is immersed in the full wake of T1 (for $\gamma_{T1} = 0°$). The upstream turbine's yaw angle is then systematically varied at nine different values $\gamma_{T1} = [-40°, -30°, -20°, -10°, 0°, +10°, +20°, +30°, +40°]$. Moreover, the streamwise separation distance between the turbines is varied from x/D=3 to x/D=6. Finally, the inflow turbulence intensity is varied from $TI_A = 0.23\%$ (Inflow A) to $TI_B = 10.0\%$ (Inflow B).





In a second test case, the effect of the lateral offset position $\Delta z/D$ of the downstream turbine T2 in the wake of an upstream turbine T1 is investigated. That means that T2 is in most cases exposed to partial wake situations. For this purpose, the lateral offset is set to seven different positions ranging from $\Delta z/D = [-0.50, -0.33, -0.16, 0, +0.16, +0.33, +0.50]$. This is done for two upstream turbine yaw angles $\gamma_{T1} = 0°$ and $\gamma_{T1} = +30°$. The turbine separation distance is kept constant at $x/D = 3$ and

only the highly turbulent inflow condition (Inflow B) is investigated.

In a third and final test case the downstream turbine yaw angle $\gamma_{T2}$ is varied as an additional parameter while it is operated at different lateral offset positions $\Delta z/D$. This concept intends to demonstrate the possibility for yaw moment mitigation in partial wake situations by opposed yawing of the downstream turbine. In this test case T2 is therefore operated at 13 different yaw angles ranging from $\gamma_{T2} = [-30°, ..., +30°]$. An overview of all investigated test cases is presented in Table 1.

For all test cases the power coefficient $C_P$, thrust coefficient $C_T$ and normalized yaw moment $M_y^*$ are assessed on T1 and T2. The power coefficient is the measured mechanical power normalized with the kinetic power of the wind in a streamtube of the same diameter:

$$C_P = \frac{P}{1/8\,\rho\,\pi\,D^2\,U_{ref}^3}. \tag{1}$$

The thrust coefficient is defined as the thrust force normal to the rotor plane normalized with the momentum of the wind in a

streamtube:

$$C_T = \frac{F_T}{1/8\,\rho\,\pi\,D^2\,U_{ref}^2}. \tag{2}$$

The yaw moment $M_y$ is normalized in a similar way as the thrust force with an additional rotor diameter $D$ to account for the normalization of the yaw moment's lever:

$$M_y^* = \frac{M_y}{1/8\,\rho\,\pi\,D^3\,U_{ref}^2}. \tag{3}$$

**3   Results**

**3.1   Operating characteristics of T1**

At first the yaw-angle dependent operating characteristics of the upstream wind turbine are presented for two inflow conditions in Figure 1. The model turbine is operated at a tip speed ratio of $\lambda_{T1} = 6.0$ for all yaw angles. The downstream turbine shows the exactly same operating characteristics when operated in undisturbed inflow. For measurements showing the power and

thrust coefficient depending on the tip speed ratio $\lambda_{T1}$ it is referred to Bartl et al. (2018).

At $\gamma_{T1} = 0$ the upstream turbine reaches a power coefficient of about $C_{P,T1} = 0.460$ for both inflow conditions. It is observed that an increase in inflow turbulence results in the same performance characteristics. As discussed by Bartl et al. (2018), the decrease in power coefficient can be approximated $C_{P,\gamma_{T1}=0} \cdot cos^3(\gamma_{T1})$ when the turbine yaw angle is varied. The thrust coefficient's reduction through yawing is observed to match well with $C_{T,\gamma_{T1}=0} \cdot cos^2(\gamma_{T1})$. The normalized yaw moment





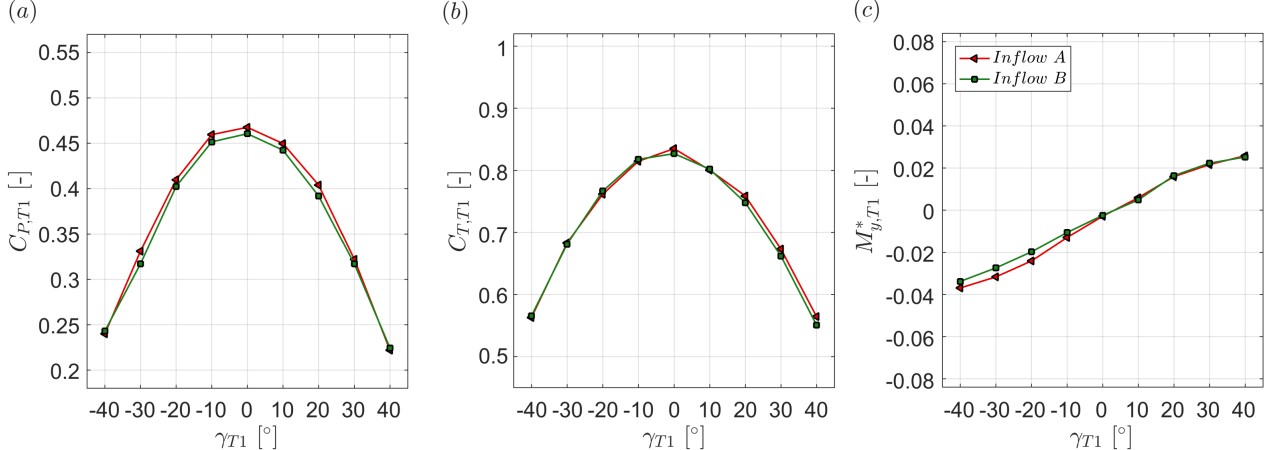

**Figure 1. (a)** Power coefficient $C_{P,T1}$ **(b)** thrust coefficient $C_{T,T1}$ and **(c)** normalized yaw moment $M^*_{y,T1}$ of the undisturbed upstream turbine T1 for different inflow conditions. The turbine is operated at $\lambda_{opt,T1} = 6.0$ for all yaw angles.

shows an almost linear behavior around the origin. However, minor asymmetries between positive and corresponding negative yaw angles are observed. These asymmetries are slightly stronger for inflow A ($TI_A = 0.23\%$).

### 3.2 Test case 1: Aligned turbines

In the first test case both rotors are installed in the center of the wind tunnel at $(y,z) = (0,0)$ aligned with the main inflow
direction. The downstream turbine position is varied from $x/D = 3$ to $x/D = 6$, while the upstream turbine yaw angle is systematically changed in steps of $\Delta\gamma_{T1} = 10°$ from $\gamma_{T1} = [-40°,...,+40°]$. Figure 2 shows two example cases, in which the downstream turbine is operated in the upstream turbine's wake for $\gamma_{T1} = 0°$ and $\gamma_{T1} = 30°$. The sketched wake flow contours in the $xz$-plane at hub height are Laser Doppler Anemometry (LDA) measurements of an example case and are only included for illustrative purposes. An exact quantification of the wake can be obtained from cross-sectional measurements in
the $yz$-plane as presented in Bartl et al. (2018). The results for the downstream turbine $C_{P,T2}$, $C_{T,T2}$ and $M^*_{y,T2}$ at inflow B in dependency of its tip speed ratio $\lambda_{T2}$ are shown in Figure 3. The downstream turbine's power is observed to increase with an increasing absolute value of the upstream turbine yaw angle. As the wake is laterally deflected, the downstream turbine is partly exposed to higher flow velocities in the freestream. The power recovery of the downstream turbine is observed to be asymmetric with respect to the upstream turbine yaw angle. Higher downstream turbine power coefficients are measured for
negative upstream turbine yaw angles. Obviously, the optimum downstream turbine T2's operating point shifts to higher tip speed ratios $\lambda_{T2}$ the more kinetic energy is available in the wake. A corresponding asymmetry between positive and negative upstream turbine yaw angles is also observed in T2's thrust coefficient, showing higher values for negative upstream turbine yaw angles. The yaw moments experienced by the downstream turbine are observed to grow with increasing upstream turbine yaw angle. As expected, downstream turbine yaw moments are positive for positive upstream turbine yaw angles and vice
versa. For low tip speed ratios, i.e. during stall the yaw moments are seen to be small and below 0.01. As soon as the flow





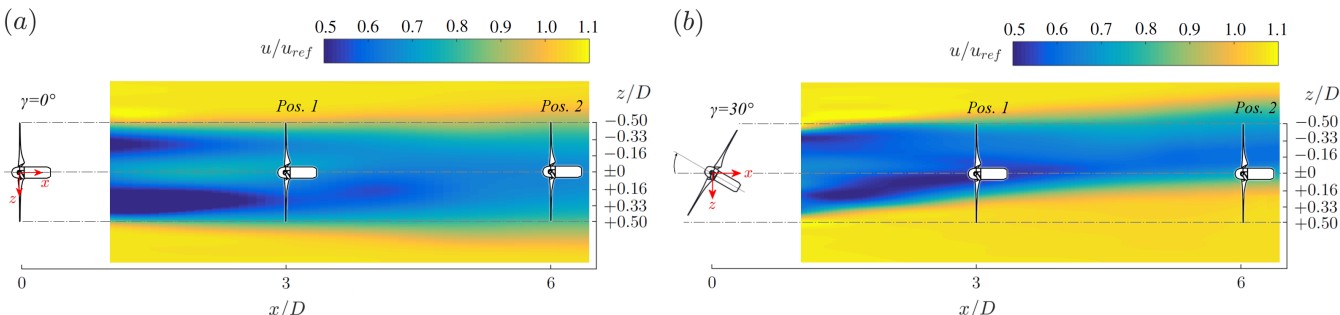

**Figure 2.** Topview of the aligned downstream turbine operated in the wake of an upstream turbine at the two different positions $x/D = 3$ and $x/D = 6$. The wake flow is indicated by measured example cases for **(a)** $\gamma_{T1} = 0°$ and **(b)** $\gamma_{T1} = 30°$.

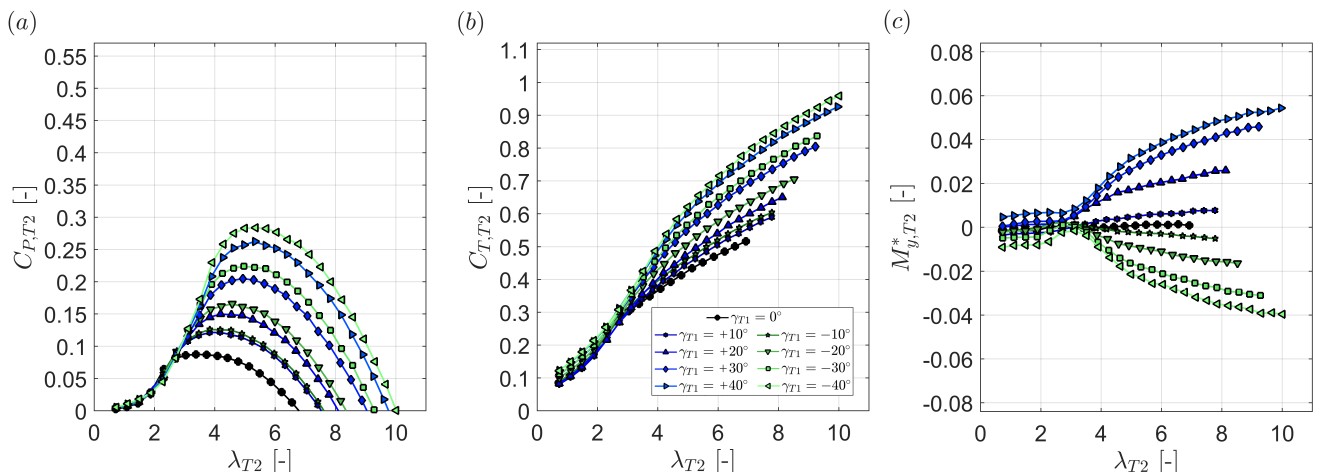

**Figure 3.** Downstream turbine **(a)** power coefficient, **(b)** thrust coefficient and **(c)** normalized yaw moment as a function of its tip speed ratio $\lambda_{T2}$ for different upstream turbine yaw angles $\gamma_{T1}$. The downstream turbine T2 is located at $x/D = 3$. The turbines are exposed to inflow B.

is attached the absolute value of the yaw moments is observed to strongly rise. Again, an asymmetry between negative and positive upstream turbine yaw angles is observed. The asymmetric wake deflection is considered to be the main reason for the asymmetric distribution of T2's yaw moments.

The effect of a variation in inflow turbulence level ($TI_A = 0.23\%$ versus $TI_B = 10.0\%$) on the downstream turbine's $C_{P,T2}$, 5  $C_{T,T2}$ and $M^*_{y,T2}$ is shown in Figure 4. The results are presented for varying upstream turbine yaw angle $\gamma_{T1}$. The downstream turbine T2 is operated at a $\lambda_{T2}$, for which $C_{P,T2}$ was maximum for the specific conditions. Note that for $x/D = 6$ neither thrust nor yaw moments were measured.

The downstream turbine's power coefficient $C_{P,T2}$ is in general observed to be higher for a higher inflow turbulence (Inflow B). The wake flow recovers at a higher rate leaving more kinetic energy for the downstream turbine to extract. The difference in 10  T2's power extraction between the two inflow turbulence levels is observed to be highest at small upstream turbine yaw angles



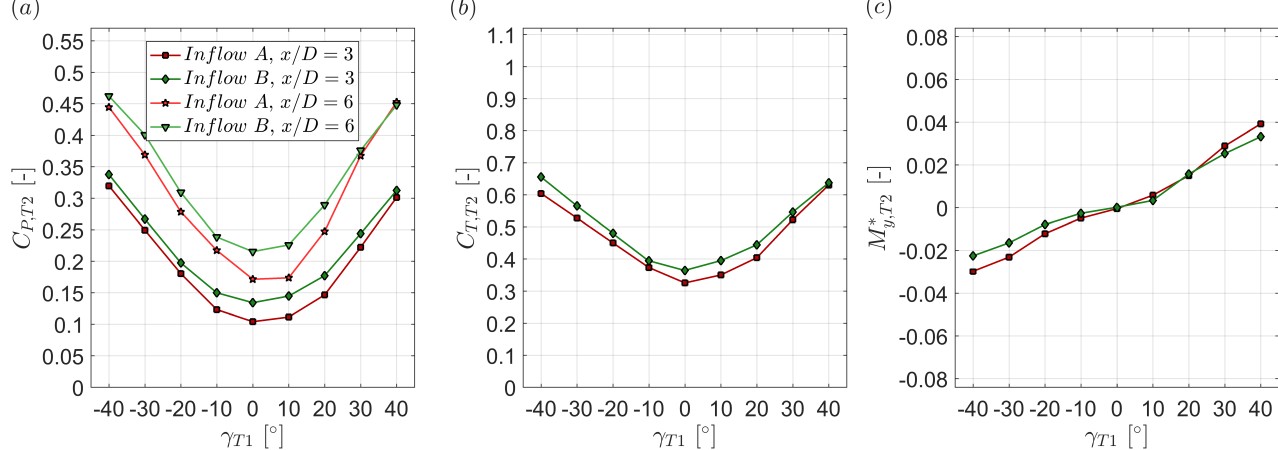

**Figure 4.** Downstream turbine **(a)** power coefficient, **(b)** thrust coefficient and **(c)** normalized yaw moment as a function of the upstream turbine's yaw angle $\gamma_{T1}$. The downstream turbine T2 is located at $x/D = 3$ and $x/D = 6$ respectively. The turbines are exposed to inflows A and B.

$\gamma_{T1}$. At high yaw angles $\gamma_{T1} \geq 30°$, however, the power coefficient $C_{P,T2}$ is very similar for the two different inflow turbulence levels. For these high yaw angles the wake's mean velocity deficit has the largest lateral deflection, exposing about half of T2's rotor swept area to the freestream. The kinetic energy content in the freestream is about the same for both inflows, which brings T2's power levels closer together. Moreover, the downstream turbine's power output at low inflow turbulence (inflow A)

is observed to be more asymmetric with respect to $\gamma_{T1}$ than at high inflow turbulence (B). Especially for $x/D = 6$, the downstream turbine power $C_{P,T2}$ is strongly asymmetric for inflow A. For extreme yaw angles $\gamma_{T1} = \pm 40°$, T2's power coefficient reaches levels of $C_{P,T2} = 0.45 - 0.46$, which is about the same magnitude of $C_{P,T1}$ at $\gamma_{T1} = 0°$. Although a considerable part of the downstream turbine rotor is impinged by T1's wake, blockage-increase freestream velocity levels of $\overline{u}/u_{ref} = 1.10$ lift the downstream turbine's power to these levels.

Similar trends are observed for the downstream turbine thrust coefficient $C_{P,T2}$ (Figure 4 (b)), where higher thrust forces are measured for the higher turbulence level in Inflow B. Inflow A implicates a higher asymmetry in $C_{T,T2}$ with respect to $\gamma_{T1}$. As previously discussed, the downstream turbine yaw moments $M_{y,T2}^*$ are observed to increase with larger upstream turbine yaw angles $\gamma_{T1}$. For both inflow cases, the yaw moments' absolute values are seen to be higher for positive $\gamma_{T1}$ than for negative $\gamma_{T1}$. Larger yaw moments are measured for Inflow A than for Inflow B, which possibly stems from stronger mean velocity

gradients in the wake flow in Inflow A. The yaw moments $M_{y,T2}^*$ on the downstream turbine located at $x/D = 3$ have approximately the same magnitude as the yaw moments measured on the upstream turbine $M_{y,T1}^*$. Consequently, an intentional upstream turbine yaw misalignment implicates significant yaw moments on the upstream turbine it self as well as an aligned downstream turbine.




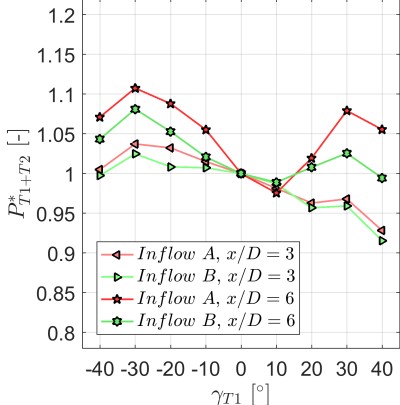

**Figure 5.** Combined relative power $P^*_{T1+T2}$ of two turbines for different upstream turbine yaw angles $\gamma_{T1}$. The downstream turbine T2 is located at $x/D = 3$ and $x/D = 6$ respectively. The turbines are exposed to inflows A and B.

A main goal of this study is to find out if upstream turbine yawing can positively affect the total power output. As observed in Figure 1 yawing the upstream turbine reduces its power output, while Figure 4 shows that the downstream turbine's power increases simultaneously. In order to quantify if the gain in T2 power can make up for the losses in T1, we define the combined relative power output of the two turbine array

$$P^*_{T1+T2} = \frac{P_{T1}(\gamma_{T1}) + P_{T2}(\gamma_{T1})}{P_{T1,\gamma_{T1}=0} + P_{T2,\gamma_{T1}=0}}. \tag{4}$$

The results for the combined relative power are presented in Figure 5 for both inflow conditions and two turbine separation distances. In all of these four setups an increase in combined power between 3.5% and 11% was measured for upstream turbine yawing. For both turbine spacings, the maximum combined efficiencies were measured for $\gamma_{T1} = -30°$. The combination of a larger wake deflection and a progressed wake recovery at higher separation distances are seen to shift the optimum of the energy balance between T1 and T2 to higher yaw angles $\gamma_{T1}$. Moreover, the combined relative power is seen to be asymmetric with higher values for negative yaw angles $\gamma_{T1}$. Both, upstream turbine power $C_{P,T1}$ and downstream turbine power $C_{P,T2}$ have seen not to be perfectly symmetrical, the larger portion can however be subscribed to the power extraction of downstream turbine exposed to asymmetric wake flow fields for positive and negative yaw angles. Furthermore, the relative power gains are observed to be significantly larger for lower inflow turbulence levels (Inflow A). Relative power gains of about 11% were measured at Inflow A, while only 8% were obtained for Inflow B at the same yaw angle of $\gamma_{T1} = -30°$.

### 3.3 Test case 2: Offset turbines

The power and loads of the downstream turbine T2 are dependent on many different parameters, such as the inflow conditions, the operating point of the upstream turbine T1, its relative streamwise and lateral position with respect to T1 as well as its operating point. In a second test case we therefore investigate the downstream turbine's performance in lateral offset. That





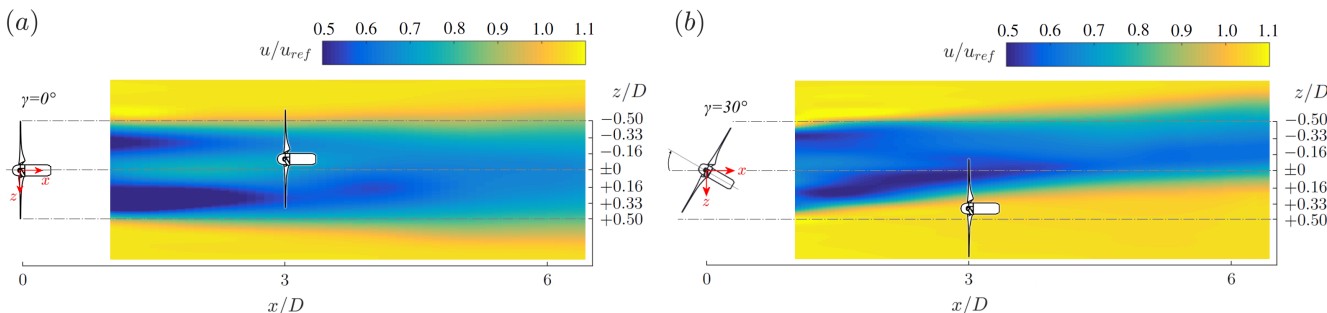

**Figure 6.** Topview of two lateral offset positions (**(a)** $\Delta z/D = -0.16$ and **(b)** $\Delta z/D = +0.33$) of the downstream turbine while operated in the wake of an upstream turbine at $x/D = 3$. The upstream turbine is operated at **(a)** $\gamma_{T1} = 0°$ and **(b)** $\gamma_{T1} = 30°$.

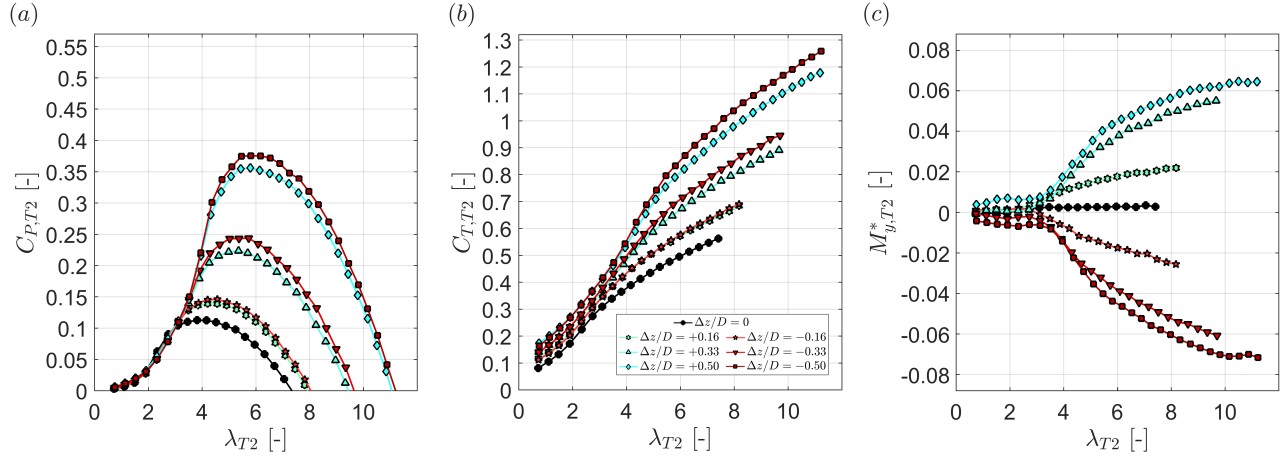

**Figure 7.** Downstream turbine **(a)** power coefficient, **(b)** thrust coefficient and **(c)** normalized yaw moment as a function of its tip speed ratio $\lambda_{T2}$ for different lateral offset positions $\Delta z/D$. The upstream turbine yaw angle is kept constant at $\gamma_{T1} = 0°$. The downstream turbine T2 is located at $x/D = 3$. The turbines are exposed to inflow B.

means that T2 experiences partial wake situations. The turbine separation distance is in this test case fixed to $x/D = 3$, while different offset positions $\Delta z/D = [-0.50, -0.33, -0.16, \pm 0, +0.16, +0.33, +0.50]$ are investigated. This is done for Inflow B ($TI_B = 10.0\%$) only, while upstream turbine yaw angles of $\gamma_{T1} = 0°$ and $\gamma_{T1} = +30°$ are investigated. In Figure 6 two example positions of the downstream turbine are sketched, illustrating two different wake impingement situations.

5   Figure 7 shows the downstream turbine's $C_{P,T2}$, $C_{T,T2}$ and $M^*_{y,T2}$ while operated in the wake of the upstream turbine at $\gamma_{T1} = 0°$ in dependency of its tip speed ratio $\lambda_{T2}$ and lateral offset position $\Delta z/D$. As expected, the power coefficient is seen to increase with increasing lateral offset $\Delta z/D$ as the downstream turbine is partly exposed to a flow of higher kinetic energy. T2's power coefficient is observed not to be entirely symmetric with respect to its lateral position in the wake. Slightly higher power coefficients are measured for negative offset positions. The reason for this is deemed to be a not perfectly axis-symmetric



velocity deficit at $x/D = 3$ as indicated in Figure 6 (a) and Bartl et al. (2018). As observed earlier, T2's optimum operating point shifts to higher tip speed ratios $\lambda_{T2}$ with increasing kinetic energy being available in the wake.

Similar trends are observed for the downstream turbine thrust coefficient $C_{T,T2}$, which was measured to be slightly higher for negative offset positions. The yaw moments experienced by the downstream turbine are seen to increase for larger lateral offsets as the rotor is impinged by stronger mean velocity gradients. The largest increases are detected for a change from $\Delta z/D = \pm 0$ to $\pm 0.16$ and from $\pm 0.16$ to $\pm 0.33$, while a position change from $\pm 0.33$ to $\pm 0.50$ only causes a small increase in yaw moment. The curves are generally observed to be almost symmetric with respect to the offset position, but also show slightly higher absolute values for negative offset positions.

The effect of a variation in upstream turbine yaw angle from $\gamma_{T1} = 0°$ to $\gamma_{T1} = 30°$ on the downstream turbine's characteristics in different lateral offset positions is presented in Figure 8. For the shown results the downstream turbine T2 is operated at a its optimum $\lambda_{T2}$, which differs for each offset position.

The red curves summarize the results for $\gamma_{T1} = 0°$ already shown in Figure 7 for their optimum operating point, while the blue curves represent a setup, in which T1 is operated at $\gamma_{T1} = 30°$ (see Figure 6). For this upstream turbine yaw angle, the wake center is shifted to $\Delta z/D = -0.167$ (Bartl et al., 2018) and correspondingly the blue curves minima in $C_{P,T2}$ and $C_{T,T2}$ are shifted to $\Delta z/D = -0.16$ (Figure 8 (a) and (b)). The yaw moment $M^*_{y,T2}$ as depicted in Figure 8 (c) is observed to be around zero for this offset position, as the rotor is approximately impinged by a full wake. For an offset position around $\Delta z/D = +0.16$ to $\Delta z/D = +0.33$ the yaw moments reach a maximum level, as roughly half the rotor swept area is impinged by the low velocity region of the wake, while the other have is impinged by the high velocity freestream flow. At a lateral offset of $\Delta z/D = +0.50$ the yaw moments on T2 are observed to decrease again. A large part of the rotor is exposed to the freestream flow; however, the wake is not yet entirely deflected away from T2. For this offset position the power and thrust coefficient are seen to reach very high levels as the rotor is exposed to a large portion of high kinetic energy freestream flow. A power coefficient of $C_{P,T2} > 0.50$ can be explained by increased freestream velocity levels of $\overline{u}/u_{ref} = 1.10$ (Bartl et al., 2018) caused by wind tunnel blockage. The power and thrust coefficient still are referred to $u_{ref}$ measured $x/D = -2$ upstream of T1.

The combined relative power output of the two-turbine array is in this case calculated for a change of upstream turbine yaw angle from $\gamma_{T1} = 0°$ to $+30°$. It has to be kept in mind, that the upstream turbine power is constant, independent of the downstream turbine position. The combined power for each offset position is calculated as

$$P^*_{T1+T2} = \frac{P_{T1,\gamma_{T1}=30} + P_{T2,\gamma_{T1}=30}(z/D)}{P_{T1,\gamma_{T1}=0} + P_{T2,\gamma_{T1}=0}(z/D)}. \tag{5}$$

Figure 9 shows the resultant combined relative power output. For an offset position of $\Delta z/D = +0.33$ a maximum combined power increase of 13% is measured, as a major part is deflected away from the downstream rotor. Surprisingly, the relative power gains measured for an offset $\Delta z/D = +0.50$ are measured to be smaller, amounting about 6%. This can be explained by significantly larger $C_{P,T2}$-values in the non-yawed case for $\Delta z/D = +0.50$ than for $\Delta z/D = +0.33$, allowing smaller relative gains. For zero lateral offset, about 5% in combined power are lost when yawing T1 to $\gamma_{T1} = +30°$ as previously





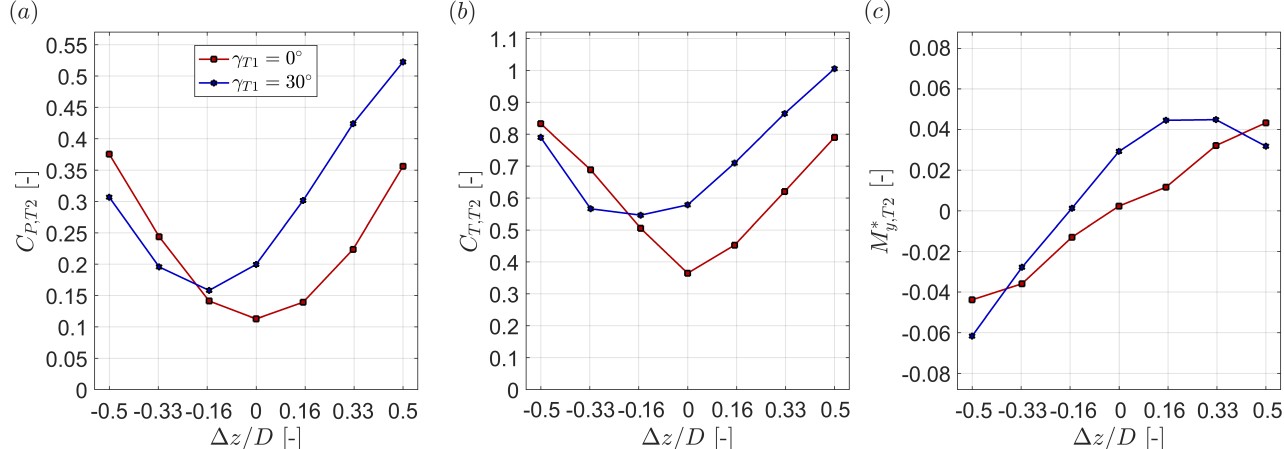

**Figure 8.** Downstream turbine **(a)** power coefficient, **(b)** thrust coefficient and **(c)** normalized yaw moment as a function of its lateral offset position $\Delta z/D$. The upstream turbine yaw angle is kept constant at $\gamma_{T1} = 0°$. The downstream turbine T2 is located at $x/D = 3$. The turbines are exposed to inflow B.

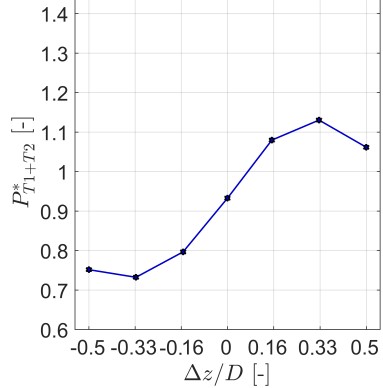

**Figure 9.** Combined relative power $P^*_{T1+T2}$ of the two-turbine-array for different lateral offset positions $\Delta z/D$. The combined power is calculated for a change of upstream turbine yaw angle from $\gamma_{T1} = 0°$ to $+30°$ for each position. The downstream turbine T2 is located at $x/D = 3$. The turbines are exposed to inflow B.

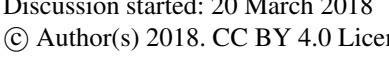
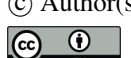


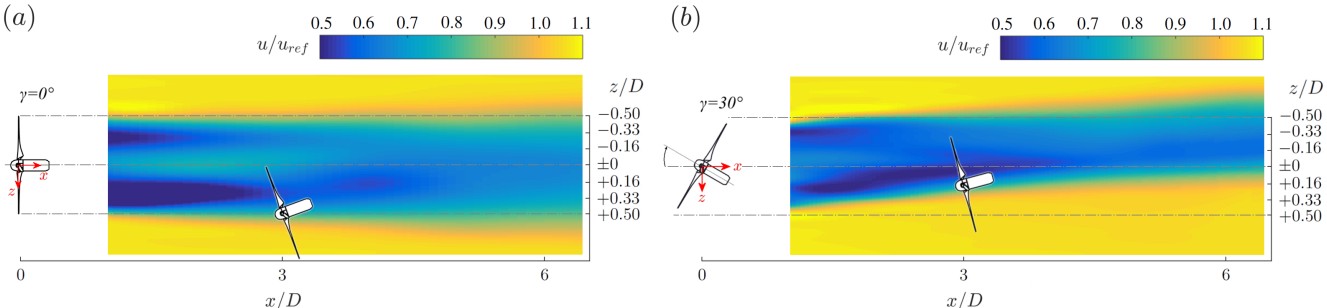

**Figure 10. (a)** Topview of the downstream turbine T2 operated at a lateral offset position $\Delta z/D = +0.50$ and a yaw angle of $\gamma_{T2} = -20°$ in the wake of an upstream turbine T1 operated at $\gamma_{T1} = 0°$. **(b)** Topview of the downstream turbine T2 operated at a lateral offset position ($\Delta z/D = +0.16$) and a yaw angle of $\gamma_{T2} = -15°$ in the wake of an upstream turbine T1 operated at $\gamma_{T1} = 30°$.

observed in Figure 5. In the case of the downstream turbine being located at negative offset positions $\Delta z/D$, the wake is deflected directly on T2's rotor, significantly reducing its power output and consequently also the combined power.

In conclusion, is has been demonstrated that intentional upstream turbine yaw control is favorable in offset situations when considering both, the power output and yaw moments on a downstream turbine. Depending on the downstream turbine's
streamwise and lateral position, the wake can be partly or even fully deflected away from its rotor swept area.

### 3.4   Test case3: Downstream turbine yawing

The third and final test case investigates whether a variation in downstream turbine yaw angle $\gamma_{T2}$ contributes to a yaw-load mitigation and power optimization. As previously seen, both partial wake impingement and turbine yaw misalignment are possible sources for increased yaw moments. An intentional yaw misalignment opposed to the partial wake impingement
is therefore considered to cancel out yaw loading on the turbine. For this purpose, the downstream turbine yaw angle is systematically varied from $\gamma_{T2}=[-30°, ..., +30°]$ in steps of $5°$ for all seven lateral offset positions and upstream turbine yaw angles $\gamma_{T1}=[0°, +30°]$. A sketch of two downstream turbine yaw angles at two offset positions is presented in Figure 10.

The resulting $C_{P,T2}$, $C_{T,T2}$ and $M^*_{y,T2}$ of the downstream turbine in dependency of its yaw angle $\gamma_{T2}$ and lateral offset position $\Delta z/D$ for a constant upstream turbine yaw angle of $\gamma_{T1} = 0°$ are shown in Figure 11. The points for $\gamma_{T2} = 0°$ cor-
respond to the previously shown red lines in Figure 8. In case the downstream turbine rotor is fully impinged by the upstream turbine's wake, i.e. $\Delta z/D = 0$, a variation of its yaw angle $\gamma_{T2}$ reduces its power output and increases uneven yaw moments. During a lateral offset however, the maximum power output and minimum yaw moments are found for yaw angles $\gamma_{T2} \neq 0°$. At a lateral offset position of $\Delta z/D = +0.16$, for instance, the maximum $C_{P,T2}$ is assessed for $\gamma_{T2} = -10°$. Simultaneously, the yaw moment is measured to be around zero at this yaw angle. The downstream turbine is exposed to a strong shear flow in
the partial wake situation, mitigating yaw moments by actively yawing opposed to that shear. The simultaneous power increase for the oppositely yawed downstream rotor is a positive side effect, although the exact reasons for the power increase are not entirely clear at this stage. Higher power outputs and decreased yaw moments are also measured for moderate yaw angles

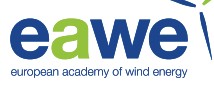
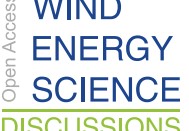


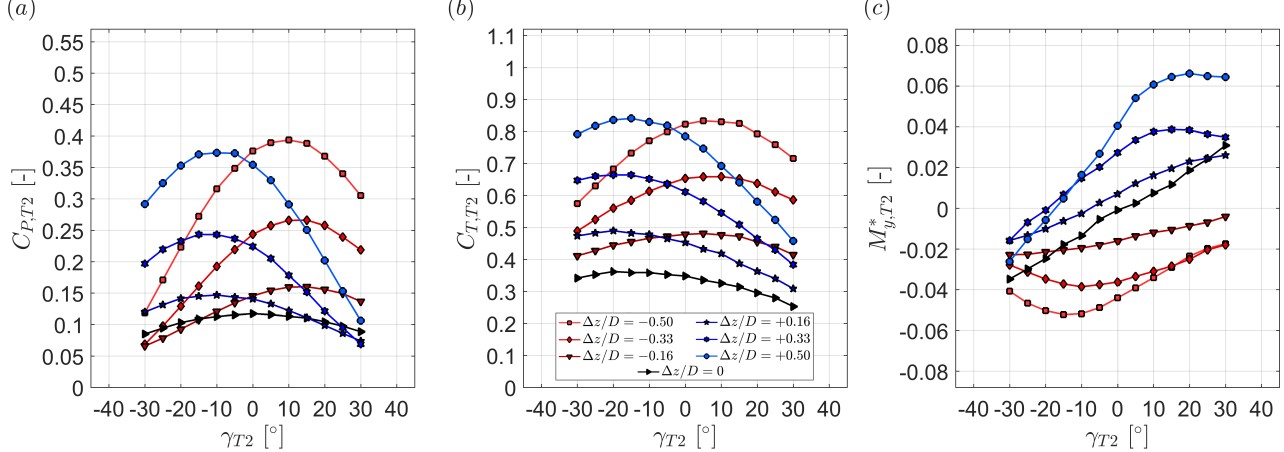

**Figure 11.** Downstream turbine **(a)** power coefficient, **(b)** thrust coefficient and **(c)** normalized yaw moment as a function of its yaw angle $\gamma_{T2}$ for different lateral offset positions $\Delta z/D$. The upstream turbine yaw angle is kept constant at $\gamma_{T1} = 0°$. The downstream turbine T2 is located at $x/D = 3$. The turbines are exposed to inflow B.

around $\gamma_{T2} = -10°$ at larger lateral offsets of $\Delta z/D = +0.33$ and $\Delta z/D = +0.50$. The slope of the power curves in Figure 11 (a) and yaw moment curves in Figure 11 (c) are observed to be even steeper for larger lateral offsets. The power gains when yawing the turbine from $\gamma_{T2} = 0°$ to $\gamma_{T2} = -10°$ are larger for higher lateral offsets. At the same time, the relative yaw moment reduction is larger, implying that opposed downstream yawing is deemed to be even more effective for higher lateral

offsets.

For negative lateral offset positions, obviously the opposite trends are observed, i.e. maximum power and smallest absolute yaw moments are measured for positive downstream turbine yaw angles $\gamma_{T2}$. The power output and yaw moment distribution is however not completely symmetrical with respect to yaw angle $\gamma_{T2}$ and offset position $\Delta z/D$.

The concept of downstream turbine yawing in partial wake impingement situations is moreover investigated for an upstream turbine yaw angle of $\gamma_{T1} = +30°$. The wake flow features a significantly higher asymmetry in this case. The results for $C_{P,T2}$, $C_{T,T2}$ and $M^*_{y,T2}$ are shown in Figure 12. As previously observed, an offset of $\Delta z/D = -0.16$ approximately corresponds to an impingement of the full wake. Thus, the power coefficient has an almost symmetric distribution with respect to downstream turbine yaw angle $\gamma_{T2}$. The yaw moments are observed to be rather low for this offset position and around zero for $\gamma_{T2} = 0$.

For partial wake impingement situations at $\Delta z/D \geq 0$, negative downstream turbine yaw angles are again seen to reduce the yaw moments acting on the rotor. The gradients in yaw moment reduction per degree of yaw angle are observed to be steeper for larger lateral offsets. The maximum power coefficients are again measured for moderate downstream turbine yaw angles around $\gamma_{T2} \pm 10°$.

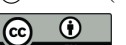

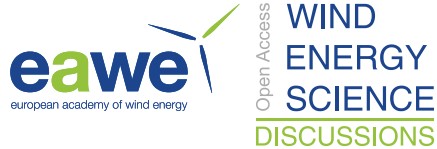

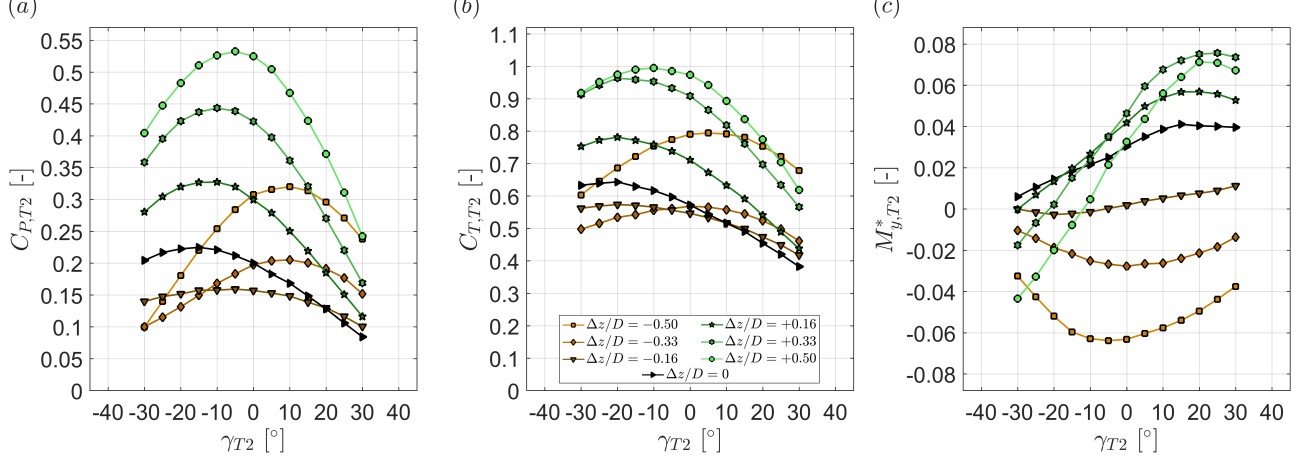

**Figure 12.** Downstream turbine **(a)** power coefficient, **(b)** thrust coefficient and **(c)** normalized yaw moment as a function of its yaw angle $\gamma_{T2}$ for different lateral offset position $\Delta z/D$. The upstream turbine yaw angle is kept constant at $\gamma_{T1} = 30°$. The downstream turbine T2 is located at $x/D = 3$. The turbines are exposed to inflow B.

Power gains by downstream turbine yawing are assessed by a relative combined power of the two-turbine array

$$P^*_{T1+T2} = \frac{P_{T1} + P_{T2}(\gamma_{T2}, z/D)}{P_{T1,\gamma_{T1}=0,z/D=0} + P_{T2,\gamma_{T1}=0,\gamma_{T2}=0,z/D=0}}. \tag{6}$$

As a reference the power measured for the non-yawed upstream turbine, a non-yawed downstream turbine in an aligned setup ($\Delta z/D = 0$) is used. The results are shown in Figure 13. For an upstream turbine yaw angle of $\gamma_{T1} = 0°$ (Figure 13 (a)) combined power gains of approximately 3% are measured for a moderate downstream turbine yaw angles ($\gamma_{T2} \pm 10 - \pm15°$). The combined power characteristics are observed to be quite symmetrical with respect to downstream turbine offset and its yaw angle. Slightly higher relative power gains are obtained for the case of an upstream turbine yaw angle of $\gamma_{T1} = +30°$ (Figure 13 (b)). A maximum power gain of about 5% is measured for offset positions $\Delta z/D = 0$ and $+0.16$ and a downstream turbine yaw angle between $\gamma_{T1} = -10°$ and $-15°$.

In conclusion, this third test case demonstrates that moderate downstream turbine yawing can be an effective method to mitigate yaw moments acting on the rotor in partial wake situations, while simultaneously obtaining slight power gains.

## 4   Discussion

When assessing the operational characteristics of the upstream turbine in dependency of its yaw angle, some asymmetries were apparent. While the power and thrust curves only showed slight deviations for positive and the corresponding negative yaw angle, higher asymmetries were found for the yaw moment. Although it is not entirely clear where these stem from, the only reasonable source for an asymmetric load distribution in an uniform inflow is the rotor's interaction with the turbine tower. In the course of a revolution, the blades of a yawed turbine experience unsteady flow conditions, i.e. fluctuations in angle of





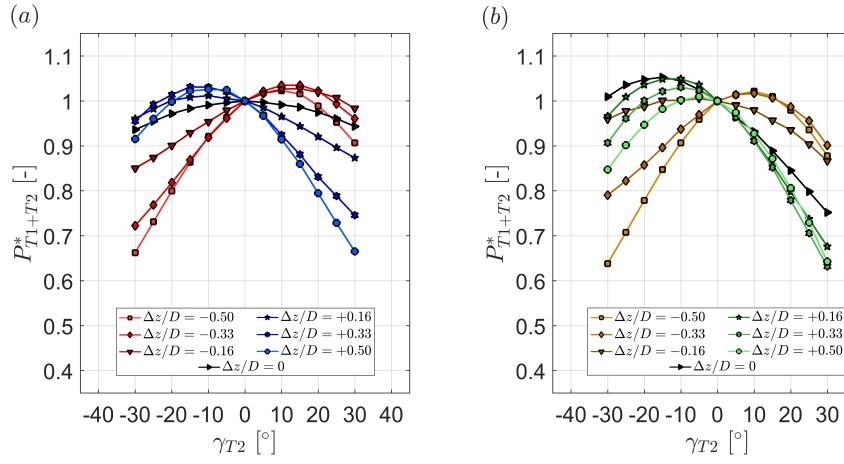

**Figure 13.** Combined relative power $P^*_{T1+T2}$ of two turbines as a function of the downstream turbine yaw angle $\gamma_{T2}$ for different lateral offset positions $\Delta z/D$. The upstream turbine yaw angle is kept constant at **(a)** $\gamma_{T1} = 0°$ and **(b)** $\gamma_{T1} = 30°$ respectively. The downstream turbine T2 is located at $x/D = 3$. The turbines are exposed to inflow B.

attack and relative velocity. When superimposing an additional low-velocity zone, tower shadow or shear for example, the yaw-symmetry is disturbed. Asymmetric load distributions for turbines exposed to sheared inflow were recently reported by Damiani et al. (2017). They showed that vertical wind shear causes asymmetric distributions of angle of attack and relative flow velocity in the course of a blade revolution. They link these to rotor loads and conclude further consequences on wake characteristics and wind farm control strategies.

Moreover, our study emphasized even stronger asymmetries in loads and power on an aligned downstream turbine. The combined power output of a two turbine setup consequently also featured an asymmetric distribution, which has been previously observed in an computational study Gebraad et al. (2016) and a similar experimental setup by Schottler et al. (2015). In a recent follow-up study, Schottler et al. (2017) attributed the asymmetry to a strong shear in the inflow to the two-turbine setup. As the inflow in the present study was measured to be spatially uniform, inflow shear is not a reason for the observed asymmetries.

The major contributor to an asymmetric combined power distribution was seen to be the downstream turbine power. The yaw angle dependency of downstream turbine power is in direct relation to an asymmetric wake deflection observed on the same setup by in Bartl et al. (2018). Therein, the wake deflection is slightly larger for negative yaw angles than for the corresponding positive yaw angles, a trend which is seen to directly affect the downstream turbine power, thrust and yaw moment distribution. The present results further demonstrate a significant influence of the inflow turbulence level on the effectiveness of wake steer-

ing by yaw. The relative power gains were observed to be significantly larger for lower inflow turbulence levels (11% versus 8%). The reason might to a small degree be differences in wake deflection (Bartl et al., 2018), but can mostly be subscribed to lower average kinetic energy levels in wakes for turbines exposed to low inflow turbulence. When deflecting a kinetic energy sink away from the downstream rotor, the relative gains in combined power are higher. Alongside with combined power increases, the results demonstrated a linear increase in the upstream turbine's yaw moments with its yaw angle. For wake



steering behind an upstream turbine, partial wake impingement situations arise for an aligned downstream turbine, resulting in increased yaw moments also on the downstream turbine.

In a real wind farm exposed to varying wind directions, however, partial wake situations, in which the downstream turbine is laterally offset are just as important as the aligned case. For a lateral offset of half a rotor diameter, for instance, it is demonstrated, that upstream turbine yaw control is able to steer most of the wake flow away from an offset downstream turbine.

Consequently, both the combined power increases and yaw moments on the downstream turbine are significantly mitigated. This finding experimentally confirms results of a similar test case recently computed with a model-framework by van Dijk et al. (2017). For an offset of $\Delta z/D = +0.33$, we measured a maximum power increase of about 13% for when yawing the upstream turbine from $\gamma_{T1} = 0°$ to $+30°$. Although not directly comparable, this result is estimated to be at the same order of magnitude as power gains experimentally obtained by Campagnolo et al. (2016), who measured a combined power increase of 21% for

a setup of three model turbines with an lateral offset of $\Delta z/D = +0.45$. Furthermore, our results indicated a not perfectly symmetrical distribution of the downstream turbine power and thrust coefficients with respect to its positive or negative offset position, as slightly higher power coefficients were obtained for negative offset positions. The reason for this is deemed to be an asymmetric velocity deficit in the non-yawed wake as indicated in Pierella and Sætran (2017) and Bartl et al. (2018).

In a final test case, we introduced the concept of downstream turbine yawing in partial wake overlap situations for the purpose

of load mitigation. The concept suggests that yawing a downstream turbine opposed to a strong horizontally sheared flow is able to mitigate rotor's yaw moments while simultaneously increasing the rotor's power output. The horizontally sheared flow is in this case the transition zone between the low- velocity wake flow to the high-velocity freestream flow. A mitigation of yaw moments by yawing the rotor opposed to the shear is intuitively imaginable, while the simultaneous power increase might be surprising. Similar effects have, however, been reported in full-scale data evaluation by McKay et al. (2013), who found an

offset in the downstream turbine's yaw alignment for the purpose of optimized power output when operated in a partial wake of an upstream turbine. The downstream turbine yaw angle was observed to adjust itself opposed to the velocity gradient in the partial wake impinging the downstream rotor. These findings are in total agreement with the optimal downstream turbine yaw angle measured in our wind tunnel experiment. The potential of load reductions of a single turbine by yawing has been previously discussed by Kragh and Hansen (2014), in situations where the rotor was exposed to vertically sheared inflows. In the

present test case, however, the partial wake impingement on the rotor represents a situation of a strongly horizontally sheared flow. Whether the shear in the incoming wind field is horizontal or vertical obviously makes a big difference, but mitigation of loads and maximization of power might be possible with yaw adjustments in both cases.

The power output and yaw moment distribution was however not completely symmetrical with respect to yaw angle $\gamma_{T2}$ and offset position $\Delta z/D$. Besides the slightly asymmetric streamwise wake flow, also the interaction of the downstream turbine

with respect to the wake rotation of the upstream turbine might cause this asymmetry. A characterization of the wake rotation and asymmetric freestream flow entrainment in the wake behind the same rotor is given by Pierella and Sætran (2017). As a yawed operation of a downstream rotor in a partial wake of an upstream turbine is highly complex, a combination of a number of different factors are assumed to influence wake-rotor interaction, making a clear conclusion difficult at this stage.



# 5 Conclusions

A wind tunnel experiment studying the effects of intentional yaw misalignment on the power production and yaw moments of a downstream turbine was presented. Both, full wake impingement and partial wake overlap situations were investigated. For partial wake overlap the concept of downstream turbine yawing for yaw moment mitigation was investigated for different lateral offset positions.

It is demonstrated that upstream turbine yaw misalignment is able to increase the combined power production of the two turbines for both partial and full wake overlap setups. For aligned turbines the combined array power was increased up to 11% for a separation distance of $x/D = 6$ and low inflow turbulence levels ($TI_A = 0.23\%$). At a higher inflow turbulence of $TI_B = 10.0\%$, however, the relative power increase was assessed to be only 8%. For smaller turbine separation distances, combined power gains were assessed to be even smaller. The distribution of combined power gains in dependency of the

upstream turbine yaw angle was observed to be rather asymmetrical. The formation of not entirely symmetric velocity deficit shapes in the wake was deemed to be the main reason for that finding.

The obtained power gains were assessed to be at the cost of increased yaw moments on the upstream rotor. The yaw moments on the upstream rotor are observed to increase roughly linearly with increasing yaw angle, but are not entirely symmetrical distributed. Upstream turbine yaw control is moreover seen to directly influence the yaw moments on a downstream rotor.

For aligned turbine positions, the downstream turbine yaw moments are observed to increase to similar magnitudes as for the upstream turbine. These results highlight the importance of also taking loads into account when optimizing layout and control of a wind farm.

Further, we demonstrate advantages of upstream turbine yaw control for load reduction and power increases on an offset downstream turbine. For situations, in which the downstream turbine is impinged by a partial wake, upstream turbine yaw

control can redirect the wake either on or away from the downstream rotor. In case the wake is directed onto the downstream turbine's rotor swept area, its yaw moments and power production reduce. If the lateral offset between the turbines is large enough, the wake can be deflected entirely away from the downstream turbine, maximizing its power and canceling out yaw moments.

Moreover, a final test case proved the concept of yaw control for yaw moment mitigation on a downstream turbine operated

in a partial wake overlap situation. While yaw moments are observed to decrease when yawing the rotor opposed to the shear layer in the incoming wake flow, also the turbine's power output is seen to increase. These results illustrate the importance for combined power and load optimization on all turbines in a wind farm.

*Competing interests.*   The authors declare that there are no competing interests.





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
