# Peer review of "Wind tunnel study on power output and yaw-moments for two yaw-controlled model wind turbines"

_Wind Energy Science, 2018_

## Referee Comment (RC1) · Anonymous Referee #1 · 18 Apr 2018

The paper 'Wind tunnel study on power and loads optimization of two yaw-controlled model wind turbines' presents interesting wind tunnel measurements of power output and yaw-moments for a two wind turbine setup, and for different intentional yaw angles, turbine spacings, and inflow conditions. The paper presents three main parametric studies. The first experiment studies the effect of yawing the upstream turbine on the power and unsteady loading of both turbines. The second experiment investigates how loading and power change for the downstream turbine, when it is moved laterally compared to the upstream turbine, and for two different yaw angles of the upstream turbine. The third experiment confirms that yaw-moments on a turbine with partial wake overlap can be mitigated by intentionally yawing in the opposite direction, and interestingly, that

this also results in a small power increase. The paper provides a clear view on how yawing and partial wake overlap can influence the yaw-moments and power output, for both the upstream and downstream turbine. This is important information for wind farm optimization and control studies. Overall, the paper is structured well, but some parts of the text can be written more clearly or improved grammatically. The reviewer has only minor comments about the scientific context.

Main comments:

- In this paper, the yaw-moment is measured as a main component for unsteady turbine loading. It would help motivate the research if the authors explain in the introduction why the yaw-moment is an important quantity.

- Figures 2,6 and 10 are confusing because they show a measured velocity plane, but the text mentions that these results should only be considered as an illustration, and are not accurate. What is the reason for this? It should be mentioned that these measurements were performed with only turbine 1. It seems indeed useful to illustrate the expected wake impact for certain turbine placements. However, it is very confusing to show measurements that are not accurate. Furthermore, if these measurements are not reliable, they cannot be used in the text to explain certain observations, see P12L1. Therefore I suggest to either provide accurate wake measurements, for instance based on the previous publication, or to draw an illustration/sketch of the expected wake and turbine placement.

- The Discussion section is too much of a repetition, and does not provide many new analyses. For example, P17L15-P18L21, do not provide any new information or observations. Therefore, the discussions seems unnecessary and more like a long conclusion. The reviewer suggests to move the few extra thoughts and references in the discussion to the corresponding parts in the main text.

- The reviewer appreciates that the control of the turbines is described clearly. The downstream turbine is controlled to its optimal performance tip-speed-ratio, for each

situations. However, the upstream turbine is controlled by keeping the tip-speed-ratio constant, even when yawed. When a turbine is yawed, it seems that the incoming velocity projected perpendicular to the rotor, decreases with the cosine of the yaw angle. By keeping the tip-speed-ratio constant to the reference velocity, one can thus expect that the yawed turbine actually operates at a relative higher tip-speed-ratio (compared to the perpendicular incoming velocity). Does this result in a less optimal performance? Because, this could mean that for a two turbine setup, with the first turbine yawed, even more optimal situations are possible with a higher aggregate power. It would be helpful if the authors discuss this in the text.

Minor comments:

- As there is no optimization in this study, it seems that the title can be made more clear by for example: 'Wind tunnel measurements of power output and yaw-moments for two yaw-controlled model wind turbines'

- Figures should be numbered according to their order of reference in the text. (figure 2 is the first to be referenced in the text)

- P4L14: In this section, it is in general not clear to which location the distances x/D refer. Is this compared to the beginning of the wind tunnel test-section? Where is the turbine located compared to the beginning of the test section?

- P17L17: '..,but can mostly by subscribed to lower average kinetic energy levels in wakes for turbines exposed to low inflow turbulence' —> This sentence doesn't provide any new information. Do the authors mean that wakes are more severe or recover more slowly when the ambient turbulence levels are lower? It is also better not to describe a wake as a kinetic energy sink, but rather as a region with low kinetic energy.

- P19L10: '.. rather asymmetrical': It could be helpful to mention other studies in the literature that also observed an asymmetrical behavior and wake deflection from yawing.

- P8L21: "Obviously, the optimum downstream turbine T2's operating point shifts to higher tip speed ratios, the more kinetic energy is available in the wake." This is not obvious to the reviewer. Maybe the authors can elaborate on the reason for this?

- P8L21: Wake recovery is not directly measured in this study. Therefore, it seems more correct to say: 'these results indicate a faster wake recovery..' + cite papers that have shown that wakes recover more quickly when turbulence levels are higher.

Technical corrections: In general, the text contains many sentences that can be written more clearly. The authors should revision the text and make it generally more easy to read. For example active vs passive voice, comma placement, and typos. See some suggestions below.

Abstract:

- 'wake overlap' instead of 'wake overlap situations'

- "For partial wake overlap the concept of downstream turbine yawing for yaw moment mitigation is examined for different lateral offset positions" —> consider splitting up this sentence to make it more easy to read.

- 'Opposed downstream turbine yawing" is not clear in the abstract. It may be more clear to say something like: "the measurements show that for a turbine with partial wake overlap, the power can be increased and the yaw moment decreased, by yawing it intentionally 10 degrees in the opposite direction."?

Main text:

- P4L12 'low turbulence' instead of 'very low'

- P4L22: keep model number as 1 part "T20W-N/2-Nm"

- Table 1: it would be helpful to indicate that yaw angles are considered from -40 to 40 in steps of 10 degrees.

- P2L 32: 'dedicated full-scale", what is meant with dedicated?

- P2L33: "They found an independent yaw alignment for the purpose of individual power increase of downstream turbines.." is not clear.

- P3L8: This is a long and complicated sentence.

- P7L13: The term 'power recovery' is not clear.

- P9L7: fix '.., blockage-increase freestream velocity levels of u/uref = 1.10 lift the downstream turbine's power to these levels.'

- P10L11: fix 'have seen not to be'

- P12L13: 'other have' is 'other halve'?

- P14L19: fix: 'The downstream turbine is exposed to a strong shear flow in the partial wake situation, mitigating yaw moment by actively yawing opposed to that shear'

- P15L4: 'deemed': 'expected' may be better?

- P15L6: remove 'obviously'
* * *

---

## Referee Comment (RC2) · Anonymous Referee #2 · 20 Apr 2018

The paper presents a wind tunnel study on the performance (both power and load) of two model wind turbines for different yaw angle distributions. Three different case studies are investigated in details: (i) the effect of the upstream turbine yaw angle under full-wake conditions, (ii) the effect of the upstream turbine yaw angle under partial-wake conditions, and (iii) yaw moment mitigation by yawing the downwind turbine.

Overall, the paper is well structured with a comprehensive literature review. The results are new and are of interest for those who study wake mitigation strategies. I have enjoyed reading the paper, and I recommend it for publication with no hesitation. I have only one major comment and a few minor comments that can be found in the

[Figure]

following:

- Figure 8(c): I found it very surprising that, for large lateral offset values such as 0.16 and 0.33 (normalized with the rotor diameter), the yaw moment of the downwind turbine is higher when the first turbine is yawed. On the contrary, I expect to see a lower moment in this case as the wake deflection essentially alleviates partial wake conditions.

- It would be useful to mention that the yaw moment can only be an indicator of unsteady loads due to inflow shear or yaw misalignment. The effect of large turbulent structures (especially those in atmospheric boundary-layer flows) on turbine loads cannot be shown by the sole consideration of yaw moment.

- I agree with the other reviewer that the discussion part is relatively redundant, and it does not add new contribution to the paper.

- Please compare your wind tunnel blockage ratio with commonly acceptable values in the literature.

- Page 6, Lines 28 and 29: Please compare your results with those reported in the literature (e.g., Ozbay et al. 2012 and Bastankhah and Porté-Agel 2017).

- Page 4, Line 17: Please add space between "... still detectable." and "At".

-Page 5, Line 15: It should be written as "... and 0.007 (0.9% of the absolute CT value), respectively".

-Figure 3: I recommend using colors with more contrast.

-Page 9, Line 9: Can it be shown using velocity measurements?

Additional references:

Ozbay, A., Tian, W., Yang, Z. and Hu, H., 2012. Interference of wind turbines with different yaw angles of the upstream wind turbine. In 42nd AIAA Fluid Dynamics Conference

and Exhibit (p. 2719).

Bastankhah, M. and Porté-Agel, F., 2017. Wind tunnel study of the wind turbine interaction with a boundary-layer flow: Upwind region, turbine performance, and wake region. Physics of Fluids, 29(6), p.065105.
* * *

---

## Referee Comment (RC3) · Anonymous Referee #3 · 10 May 2018

The paper presents valuable experimental results that could contribute to better understanding the potential of wake steering by yawing applied to wind farms. However, I see the following weak points that could be improved, as well as some minor corrections.

Page 5, Line 14-15: The sentence should be rewritten as follows: "The total uncertainties in power and thrust coefficient are 0.006 (2.5% of the absolute 15 CP -value) and 0.007 (0.9% of the absolute CT -value), respectively."

Page 8, line 2-3: "The asymmetric wake deflection is considered to be the main reason for the asymmetric distribution of T2's yaw moments.". It is quite clear that yawing the upstream wind turbine in two different direction leads to different power gain on

the downstream one. The authors trace back this behavior to not-well specified asymmetric wake deflection. It would be interesting, for the readers, if the authors could provide a deeper insight into this topic, considering that the authors (previously cited publication) already measured the wake shed by the upstream WT for two different yaw misalignment. Is the observed asymmetry due to asymmetric wake displacement or wake recovery?

Page 9, line 4-5: "Moreover, the downstream turbine's power output at low inflow turbulence (inflow A) is observed to be more asymmetric with respect to $\gamma$T1 than at high inflow turbulence (B)." This is quite surprising since one would expect that as more homogenous the flow is, as higher the symmetry of the phenomena is. It would be interesting if the authors could argue more about the reasons behind the observed data.

Page 10. In a previous sentence, the authors reported that quite substantial wake blockage was observed, leading to an increase of 10% of the speed outside the wake of the upstream model. How much is the blockage affecting the results presented in Figure 5? Moreover, the rotor speed of the upstream model was kept constant even for a very high yaw misalignment, which implies that the upstream model is operating at sub-optimal conditions. Indeed, when yawing a wind turbine it would have been better to keep constant the effective TSR, i.e. the TSR computed by using the component of the wind speed orthogonal to the rotor disk. How much power is lost, on the upstream model, due to the fact the model itself is operating, while yawed, at sub-optimal conditions? How this affects the results presented in figure 5?

Page 11, line 1: the authors claim that the lack of symmetry, in the power output, for a downstream model placed on the right or left side of the upstream one, is due to "not perfectly axis-symmetric velocity deficit at x/D = 3". Since the authors measured the wake shed by the upstream wind turbine, it would be beneficial to add also a quantitative comparison: could the measured not perfectly axis-symmetric velocity deficit quantitatively explain the observed difference of power output?

Page 12: which is the effect of wake blockage on the data reported in Figure 8? As the authors properly write, the high Cp, measured on the downstream turbine experiencing partial-wake conditions, is due to blockage. How would the plots in figure 8 look like if the effects of blockage were compensated?

Page 13, Figure 8. The caption reports: "The upstream turbine yaw angle is kept constant at gamma = 0". It should be "The downstream turbine yaw angle is kept constant at gamma = 0"

Page 16: quite surprisingly, it is found that the downstream wind turbine should be yawed by 10-15 degrees ( quite a lot!) in order to improve its power production. However, again the TSR of the second turbine was not changed while varying its misalignment angle. This could again lead to sub-optimal operating conditions. If the models were operated as full-scale wind turbines are (constant effective TSR) the conclusions could have been quite different. The authors should comment on this.
* * *

---

## Author Comment (AC1) · 8 Jun 2018

**Authors' response to Referee #1:**

We would like to thank the referee for reviewing this manuscript, the valuable feedback and the very constructive comments. At this stage of the review process, we respond to the referee #1's comments and propose improvements for the final manuscript. The referee's original comments are printed in **bold** followed by the corresponding answers. Passages from the manuscript are printed in *italic writing*, in which proposed additions are indicated in blue and deleted parts in .
Thank you very much for your efforts,

Jan Bartl on behalf of all authors
* * *
**Main comment (1)**
**In this paper, the yaw-moment is measured as a main component for unsteady turbine loading. It would help motivate the research if the authors explain in the introduction why the yaw-moment is an important quantity.**

Thank you for this very good comment. Indeed, the connection of the yaw-moment acting on a rotor to unsteady loading is not sufficiently explained in the text. We therefore suggest the following addition to the introduction in the manuscript:

p.3, l.19 f:
*For this purpose the parameters turbine separation distance $x/D$, lateral turbine offset $z/D$ and turbine yaw settings $\gamma_{T1}$ and $\gamma_{T2}$ are systematically varied in this wind tunnel experiment. Aside from power output and rotor thrust, the yaw moments acting on the individual rotors are measured. Yaw moments are a representation of the imbalance of the forces acting on a rotor blade during the course of one rotation. High values of yaw moments thus indicate increased unsteady blade loading at a frequency of the corresponding the rotational speed. Special focus is given to the concept of downstream turbine yawing (…).*

**Main comment (2)**
**Figures 2,6 and 10 are confusing because they show a measured velocity plane, but the text mentions that these results should only be considered as an illustration, and are not accurate. What is the reason for this? It should be mentioned that these measurements were performed with only turbine 1. It seems indeed useful to illustrate the expected wake impact for certain turbine placements. However, it is very confusing to show measurements that are not accurate. Furthermore, if these measurements are not reliable, they cannot be used in the text to explain certain observations, see P12L1. Therefore I suggest to either provide accurate wake measurements, for instance based on the previous publication, or to draw an illustration/sketch of the expected wake and turbine placement.**

We agree with the reviewer, that the presented velocity planes in Figures 2, 6 and 10

of the manuscript might be confusing in this context. The shown velocity planes are considered to be accurate, but were measured behind a smaller version of the original rotor ($D_{small} = 0.45m$ vs. $D_{orig.} = 0.90m$). In the previous publication (Bartl et al., 2018) the wake deflections behind these two rotors were assessed to be very similar. Thus, the portion of the wake impacting the downstream turbine as shown in the Figures is deemed to be representative for the real situation.

As we do not intent to repeat wake measurements of the previous publication, we suggest to use sketches of the expected wake and turbine placement in the final version of the manuscript as shown below. The following text passages will be modified accordingly:

p.7, l.7:
*The sketched wake flow contours in the xz-plane at hub height are  included for illustrative purposes.* *The location of the wake flow as sketched in gray is roughly estimated from previously performed measurements as presented in Bartl et al. (2018).*

p.11, l.9 and p.12. l.1:
*The reason for this is deemed to be a not perfectly axis-symmetric velocity deficit at $x/D = 3$ as indicated in  Bartl et al. (2018).*

[Figure]

Figure 1: **Figure 2.** Topview of the aligned downstream turbine operated in the wake of an upstream turbine at the two different positions $x/D = 3$ and $x/D = 6$. The wake flow is indicated  for **(a)** $\gamma_{T1} = 0°$ and **(b)** $\gamma_{T1} = 30°$.

[Figure]

Figure 2: **Figure 6.** Topview of two lateral offset positions (**(a)** $z/D = -0.16$ and **(b)** $z/D = +0.33$) of the downstream turbine while operated in the wake of an upstream turbine at $x/D = 3$. The upstream turbine is operated at **(a)** $\gamma_{T1} = 0°$ and **(b)** $\gamma_{T1} = 30°$.

[Figure]

Figure 3: **Figure 10. (a)** Topview of the downstream turbine T2 operated at a lateral offset position $z/D = +0.50$ and a yaw angle of $\gamma_{T2} = -20°$ in the wake of an upstream turbine T1 operated at $\gamma_{T1} = 0°$. **(b)** Topview of the downstream turbine T2 operated at a lateral offset position ($z/D = +0.16$) and a yaw angle of $\gamma_{T2} = -15°$ in the wake of an upstream turbine T1 operated at $\gamma_{T1} = 30°$.

**Main comment (3)**

**The Discussion section is too much of a repetition, and does not provide many new analyses. For example, P17L15-P18L21, do not provide any new information or observations. Therefore, the discussions seems unnecessary and more like a long conclusion. The reviewer suggests to move the few extra thoughts and references in the discussion to the corresponding parts in the main text.**

Thank you for this constructive comment. We agree that the discussion mainly repeats previously presented results and only sparsely provides new information. We therefore follow the reviewers suggestion to completely omit the Discussion section and move the comparisons with external sources to the results section. These references are moved to the following sections in the text:

p.7, l.3 f:
*These asymmetries are slightly stronger for inflow A ($TI_A = 0.23\%$). Although it is not entirely clear where these stem from, the only reasonable source for an asymmetric load distribution in an uniform inflow is the rotor's interaction with the turbine tower. In the course of a revolution, the blades of a yawed turbine experience unsteady flow conditions, i.e. fluctuations in angle of attack and relative velocity. When superimposing an additional low-velocity zone, tower shadow or shear for example, the yaw-symmetry is disturbed. Asymmetric load distributions for turbines exposed to sheared inflow were recently reported by Damiani et al. (2017). They showed that vertical wind shear causes asymmetric distributions of angle of attack and relative flow velocity in the course of a blade revolution. They link these to rotor loads and conclude further consequences on wake characteristics and wind farm control strategies.*

p.10, l.14 f:
*Relative power gains of about 11% were measured at Inflow A, while only 8% were obtained for Inflow B at the same yaw angle of $\gamma_{T1} = -30°$. Asymmetries in the combined power output have been previously observed in a computational study Gebraad et al. (2016) and a similar experimental setup by Schottler et al. (2015). In a recent follow-up study, Schottler et al. (2017) attributed the asymmetry to a strong shear in*

*the inflow to the two-turbine setup. As the inflow in the present study was measured to be spatially uniform, inflow shear is not a reason for the observed asymmetries.*

p.14, l.3 ff:
*In conclusion, is has been demonstrated that intentional upstream turbine yaw control is favorable in offset situations when considering both, the power output and yaw moments on a downstream turbine. Depending on the downstream turbine's streamwise and lateral position, the wake can be partly or even fully deflected away from its rotor swept area. This finding experimentally confirms results of a similar test case recently computed with a model-framework by van Dijk et al. (2017).*

p.14, l.18 f:
*Simultaneously, the yaw moment is measured to be around zero at this yaw angle. The potential of load reductions of a single turbine by yawing has been previously discussed by Kragh and Hansen (2014), in situations where the rotor was exposed to vertically sheared inflows. In the present test case, however, the partial wake impingement on the rotor represents a situation of a strongly horizontally sheared flow. Whether the shear in the incoming wind field is horizontal or vertical obviously makes a big difference, but mitigation of loads and maximization of power might be possible with yaw adjustments in both cases.*

p.14, l.20 f:
*The simultaneous power increase for the oppositely yawed downstream rotor is a positive side effect, although the exact reasons for the power increase are not entirely clear at this stage. A power increase by downstream turbine yawing has previously been reported in a full-scale data evaluation by McKay et al. (2013), who found an offset in the downstream turbine's yaw alignment for the purpose of optimized power output when operated in a partial wake of an upstream turbine. The downstream turbine yaw angle was observed to adjust itself opposed to the velocity gradient in the partial wake impinging the downstream rotor. These findings are in total agreement with the optimal downstream turbine yaw angle measured in our wind tunnel experiment.*

**Main comment (4)**
The reviewer appreciates that the control of the turbines is described clearly. The downstream turbine is controlled to its optimal performance tip-speed-ratio, for each situations. However, the upstream turbine is controlled by keeping the tip-speed-ratio constant, even when yawed. When a turbine is yawed, it seems that the incoming velocity projected perpendicular to the rotor, decreases with the cosine of the yaw angle. By keeping the tip-speed-ratio constant to the reference velocity, one can thus expect that the yawed turbine actually operates at a relative higher tip-speed-ratio (compared to the perpendicular incoming velocity). Does this result in a less optimal performance? Because, this could mean that for a two turbine setup, with the first turbine yawed, even more optimal situations are possible with a higher aggregate power. It would be helpful if the authors discuss this in the text.

This is a very good thought and indeed requires a deeper discussion in the text. We have measured the operating characteristics of the upstream turbine in dependence of the yaw angle and tip speed ratio. For $\gamma_{T1} = 0°$ and $\pm 30°$ the operating characteristics for all inflow conditions are shown in the previous publication (Bartl et al., 2018), which already is referred to in the text. The complete characteristics for $\gamma_{T1} = 0°$ to $+40°$ (Inflow B) are shown here in Figure 4 for positive yaw angles only (note that negative yaw angles have a very similar TSR-dependency). It can observed that the maximum power coefficient is measured at $\lambda = 6.0$ for yaw angles between 0° and 30°. For the highest yaw angle of 40°, however, the optimum tip speed ratio is found at $\lambda = 5.5$, which makes sense according to the reasoning given by the reviewer. At this extreme yaw angle, a slightly higher combined power output could indeed have been achieved, if the upstream turbine would have been operated at $\lambda = 5.5$. However, a constant upstream turbine tip speed ratio of $\lambda = 6.0$ seems to be optimum for the most interesting region between 0° and 30°.

Nevertheless, we suggest to add some additional lines of text to the manuscript discussing the TSR-dependency.

[Figure]

Figure 4: Tip-speed-ratio-dependent operating characteristics of the upstream turbine T1 operated at yaw angles from $\gamma_{T1} = 0°$ to $+40°$ at inflow B.

p.6, l.23 ff:
*The model turbine is operated at a tip speed ratio of $\lambda_{T1} = 6.0$ for all yaw angles.*  *For measurements showing the power and thrust coefficient depending on the tip speed ratio $\lambda_{T1}$ it is referred to Bartl et al. (2018). There, the power coefficient is assessed to be maximum at $\lambda_{T1} = 6.0$ for all yaw angles between $\gamma_{T1} = 0°$ to $\pm 30°$. A slight shift towards a lower optimum tip speed ratio of $\lambda_{T1} = 5.5$ is measured for $\gamma_{T1} = \pm 40°$ (not shown in graph). As the difference in total power coefficient is observed to be very small, the upstream turbine is constantly operated at $\lambda_{T1} = 6.0$ also for these yaw angles. The downstream turbine shows exactly the same operating characteristics when operated in undisturbed inflow.*

**Minor comment (1)**
**As there is no optimization in this study, it seems that the title can be made more clear by for example: 'Wind tunnel measurements of power output and yaw-moments for two yaw-controlled model wind turbines'**

We agree that the term "optimization" does not reflect the content of this study, and therefore should be excluded from the title. We suggest to use a mixture of the reviewer's suggestion and the original title: "Wind tunnel study on power output and yaw-moments for two yaw-controlled model wind turbines"

p.1, l.0 (Title):

_Wind tunnel study on power output and yaw-moments for two yaw-controlled model wind turbines_

**Minor comment (2)**
**Figures should be numbered according to their order of reference in the text. (figure 2 is the first to be referenced in the text).**

Thank you for the hint. This line was obviously added in a revision of the text, violating the correct order. We therefore suggest to move this line to a later location in the text.

p.4, l.1:
_(...) model wind turbines rotate counter-clockwise._

p.7, l.6 f:
_Figure 2 shows two example cases, in which the downstream turbine is operated in the upstream turbine's wake for $\gamma_{T1} = 0°$ and $\gamma_{T1} = 30°$._ _Positive yaw is defined as indicated in Figure 2._

**Minor comment (3)**
**P4L14: In this section, it is in general not clear to which location the distances x/D refer. Is this compared to the beginning of the wind tunnel test-section? Where is the turbine located compared to the beginning of the test section?**

We agree that this is not well explained in the text. $x/D = 0$ refers to the location of the upstream turbine, which is not clear before studying the sketches in Figure 2. In order to make this clearer, we suggest to make a small addition to the text:

p.4, l.13 f:
_Inflow B is generated by a static grid at the wind tunnel inlet $(x/D = -2)$ and is measured to amount $TI_B = 10.0\%$ at the location of the upstream turbine $(x/D = 0)$._

*The grid-generated turbulence decays with increasing downstream distance to about $TI_B$ = 5.5% at $x/D = 3$ and to $TI_B = 4.0\%$ at $x/D = 6$.*

**Minor comment (4)**

**P17L17: '..,but can mostly by subscribed to lower average kinetic energy levels in wakes for turbines exposed to low inflow turbulence. This sentence doesn't provide any new information. Do the authors mean that wakes are more severe or recover more slowly when the ambient turbulence levels are lower? It is also better not to describe a wake as a kinetic energy sink, but rather as a region with low kinetic energy.**

We agree that the sentence does not provide any new useful information. As already discussed in Major comment (3), the Discussion section is suggested to be omitted in the final version of the manuscript (with single comparisons being moved to the Results section).

Yes, the reviewer's interpretation of the sentence's meaning is correct, but that has already been discussed earlier in the text.

**Minor comment (5)**

**'.. rather asymmetrical': It could be helpful to mention other studies in the literature that also observed an asymmetrical behavior and wake deflection from yawing.**

We have now moved two references, which also observed asymmetries in the combined power output, from the Discussion section to the results section. Thus, this finding is now directly discussed in the text.

p.10, l.14 f:

*Relative power gains of about 11% were measured at Inflow A, while only 8% were obtained for Inflow B at the same yaw angle of $\gamma_{T1} = -30°$. Asymmetries in the combined power output have been previously observed in a computational study Gebraad et al. (2016) and a similar experimental setup by Schottler et al. (2015). In a recent follow-up study, Schottler et al. (2017) attributed the asymmetry to a strong shear in the inflow to the two-turbine setup. As the inflow in the present study was measured to be spatially uniform, inflow shear is not a reason for the observed asymmetries.*

**Minor comment (6)**

**P8L21: "Obviously, the optimum downstream turbine T2's operating point shifts to higher tip speed ratios, the more kinetic energy is available in the wake." This is not obvious to the reviewer. Maybe the authors can elaborate on the reason for this?**

Thank you for the comment. This is indeed not sufficiently explained in the text yet. The reason for higher optimum tip speed ratios of the downstream turbine is the fact, that also the power coefficient $C_{P,T2}$ is referred to the constant far upstream reference

velocity $U_{ref}$ and not the local inflow velocity to the downstream turbine (which is difficult to define due to its spatial non-uniformity). We therefore suggest to add two short sentences; one where we define the power, thrust and yaw moment coefficients and the other in the discussion of the results, respectively.

p.6, l.10:
*For all test cases the power coefficient $C_P$ , thrust coefficient $C_T$ and normalized yaw moment $M_y^*$ are assessed on T1 and T2. Note that the coefficients for both turbines are normalized with the reference inflow velocity $U_{ref}$ measured far upstream of the turbine array at $x/D = -2$.*

p.7, l.15 f:
*Obviously, the The optimum downstream turbine T2's operating point shifts to higher tip speed ratios $\lambda_{T2}$, the more kinetic energy is available in the wake. As the downstream turbine power coefficient is referred to the constant far upstream reference velocity $U_{ref}$, the optimum operating conditions are measured for higher tip speed ratios as soon as the local inflow velocity increases.*

**Minor comment (7)**

**P8L21: Wake recovery is not directly measured in this study. Therefore, it seems more correct to say: 'these results indicate a faster wake recovery..' + cite papers that have shown that wakes recover more quickly when turbulence levels are higher.**

We not completely sure, if we are looking at the same sentence in the text here, as there is no P8L21 in the manuscript. Referring to P8L9, we agree that this is not a result of the presented study, but rather the previous wake study (Bartl et al., 2018). We therefore suggest to add a reference here.

p.8, l.9:
*As previously observed in Bartl et al. (2018), the The wake flow recovers at a higher rate, leaving more kinetic energy for the downstream turbine to extract.*

**Technical correction (1)**
**Abstract: - 'wake overlap' instead of 'wake overlap situations'.**

Thank you for pointing out a number of technical mistakes. All of them will be included in the final version of the manuscript, in order to make the text easier to read.

p.1, l.1 f:
*In this experimental wind tunnel study the effects of intentional yaw misalignment on the power production and loads of a downstream turbine are investigated for full and partial wake overlap .*

**Technical correction (2)**
**Abstract: - "For partial wake overlap the concept of downstream turbine yawing for yaw moment mitigation is examined for different lateral offset positions." - consider splitting up this sentence to make it more easy to read.**

The referred sentence is actually from the conclusions. But an even longer, more complicated sentence is found in the abstract. We agree that both sentences are too long and complicated. We suggest to split up the abtract's sentence and to omit the second part of the conclusion's sentence:

p.1, l.9 ff:
*For partial wake overlap , yaw moments on the downstream turbine can be mitigated through upstream turbine yawing . Simultaneously, the combined power output of the turbine array is increased.*

p.19, l.3 f:
*For partial wake overlap the concept of downstream turbine yawing for the purpose of yaw-moment mitigation is examined .*

**Technical correction (3)**
**Abstract: - "Opposed downstream turbine yawing" is not clear in the abstract. It may be more clear to say something like: "the measurements show that for a turbine with partial wake overlap, the power can be increased and the yaw moment decreased, by yawing it intentionally 10 degrees in the opposite direction."?**

We agree that this concept of "opposed downstream turbine yawing" is not yet introduced, and therefore not suited in the abstract. We suggest the following wording:

p.1, l.11 f:
*A final test case demonstrates  benefits for power and loads through downstream turbine yawing in partial wake *

*reduce its yaw moments and increasing its power production by up to 5%.*overlap. *Yaw moments can be decreased and the power increased by intentionally yawing the downstream turbine in the opposite direction.*

**Technical correction (4)**
**Main text: - P4L12 'low turbulence' instead of 'very low'**

We agree.

p.4, l.12:
*(...) an inflow of  low turbulence intensity (...)*

**Technical correction (5)**
**Main text: - P4L22: keep model number as 1 part "T20W-N/2-Nm".**

Yes. Thank you for the hint.

p.17, l.18 ff:
*(...) HBM torque transducer of the type "T20W-N/2-Nm".*

**Technical correction (6)**
**Main text: - Table 1: it would be helpful to indicate that yaw angles are considered from -40 to 40 in steps of 10 degrees.**

We will add an additional number to indicate the steps of 10 degrees.

Table 1:
$[-40°, -30°..., +40°]$

**Technical correction (7)**
**Main text: - P2L 32: "dedicated full-scale", what is meant with dedicated?**

The wording is probably not well chosen here. We suggest to use "comprehensive" instead of "dedicated" here.

p.2, l.32:
*A comprehensive full-scale study by McKay et al. (2013) (...)*

**Technical correction (8)**
**Main text: - P2L33: "They found an independent yaw alignment for the purpose of individual power increase of downstream turbines.." is not clear.**

We agree that this sentence is not clear at all. We suggest a new wording and sentence

structure:

p.2, l.33 f:
*They found*  _a power increase for_ *downstream turbines,* _which independently misaligned their yaw angle from the main wind direction when_ *operated in partial wake situations.*

**Technical correction (9)**
**Main text: - P3L8: This is a long and complicated sentence.**

We agree and suggest to shorten down the sentence by deleting needless parts of it.

p.3, l.8 ff:
*In a computational setup of ten aligned*  *turbines, Andersen et al. (2017)*  *investigated the influence of inflow* _conditions_  *and*  *turbine spacing on*  *yaw moments*  _of_ *downstream turbines operated in the wake.*

**Technical correction (10)**
**Main text: - P7L13: The term 'power recovery' is not clear.**

This is indeed not clear. We suggest to use the word "output" instead.

p.7, l.13 ff:
*The power*  _output_ *of the downstream turbine is observed to be asymmetric with respect to the upstream turbine yaw angle.*

**Technical correction (11)**
**Main text: - P9L7: fix '.., blockage-increase freestream velocity levels of u/uref = 1.10 lift the downstream turbine's power to these levels.'**

We agree, that this is again not very well-explained. We consider a full revision of this sentence, adding a deeper explanation of the assumed effects.

p.9, l.7 ff:
_These high downstream power coefficients $C_{P,T2}$ can be explained by increased velocity levels of $u/u_{ref} = 1.10$ in the freestream outside of the wake as a result of wind tunnel blockage (Bartl et al., 2018). The downstream turbine power coefficient is, however, still referred to the undisturbed far upstream reference velocity $u_{ref}$._ *Although a considerable part of the downstream turbine rotor is impinged by T1's wake,*  _higher wind speeds outside of the wake_ *lift the downstream turbine's power to these levels.*

**Technical correction (12)**
**Main text: - P10L11: fix 'have seen not to be'**

This is indeed bad language and will be fixed in the manuscript. Also, the rather long sentence is split up into two parts.

p.10, l.11 ff:
*Both, upstream turbine power $C_{P,T1}$ and downstream turbine power $C_{P,T2}$*  *are observed to be asymmetrically distributed. The larger portion can however be subscribed to the power extraction of downstream turbine, which is exposed to asymmetric wake flow fields for positive and negative yaw angles.*

**Technical correction (13)**
**Main text: - P12L13: 'other have' is 'other halve'?**

This is indeed a typing mistake. We suggest to omit the second part of the sentence, as it only makes the sentence unnecessarily long.

p.12, l.16 ff:
*For an offset position around $z/D = +0.16$ to $z/D = +0.33$ the yaw moments reach a maximum level, as roughly half the rotor swept area is impinged by the low velocity region of the wake*  *.*

**Technical correction (14)**
**Main text: - P14L19: fix: 'The downstream turbine is exposed to a strong shear flow in the partial wake situation, mitigating yaw moment by actively yawing opposed to that shear'.**

We agree that this sentence grammatically does not make any sense. We suggest the following correction:

p.14, l.19 f:
*As the*  *downstream turbine operated in the partial wake*  *exposed to a strongly sheared inflow,*  *yaw moments can be mitigated by actively yawing the rotor in the opposite direction to the incoming*  *shear.*

**Technical correction (15)**
**Main text: - P15L4: 'deemed': 'expected' may be better?**

We agree and pick up the suggested correction.

p.15, l.4 ff:
*At the same time, the relative yaw moment reduction is larger, implying that opposed*

*downstream yawing is*  *expected to be even more effective for higher lateral offsets.*

**Technical correction (16)**
**Main text: - P15L6: remove 'obviously'.**

We agree that 'obviously' does not fit here.

p.15, l.6 f:
*For negative lateral offset positions,*  *the opposite trends are observed, i.e. maximum power and smallest absolute yaw moments are measured for positive down-stream turbine yaw angles.*

**References**

[1] Bartl, J., Mühle, F., Schottler, J., Hölling, M., Peinke, J., Adaramola, M., and Sætran, L.: Wind tunnel experiments on wind turbine wakes in yaw: Influence of inflow turbulence and shear, Wind Energ. Sci., 3, 329–343, doi:10.5194/wes-3-329-2018, 2018.

[2] Damiani, R., Dana, S., Annoni, J., Fleming, P., Roadman, J., van Dam, J., and Dykes, K.: Assessment of Wind Turbine Component Loads Under Yaw-Offset Conditions, Wind Energ. Sci., 3, 173–189, doi:10.5194/wes-3-173-2018, 2018.

[3] Gebraad, P. M. O., Teeuwisse, F. W., van Wingerden, J. W., Fleming, P. A., Ruben, S. D., Marden, J. R., and Pao, L. Y.: Wind plant power optimization through yaw control using a parametric model for wake effects-a CFD simulation study, Wind Energy, 19, 95–114, doi:10.1002/we.1822, 2016.

[4] Schottler, J., Hölling, A., Peinke, J., and Hölling, M.: Wind tunnel tests on controllable model wind turbines in yaw, AIAA 34thWind Energy Symposium, 1523, doi:10.2514/6.2016-1523, 2015.

[5] Schottler, J., Hölling, A., Peinke, J., and Hölling, M.: Brief communication: On the influence of vertical wind shear on the combined power output of two model wind turbines in yaw, Wind Energy Science, 2, 439–442, doi: 10.5194/wes-2-439-2017, 2017a.

[6] van Dijk, M., van Wingerden, J.-W., Ashuri, T., and Li, Y.: Wind farm multi-objective wake redirection for optimizing power production and loads, Energy, 121, 561–569, doi:10.1016/j.energy.2017.01.051, 2017.

[7] Kragh, K. A. and Hansen, M. H.: Load alleviation of wind turbines by yaw misalignment, Wind Energy, 17, 971–982, doi:10.1002/we.1612, 2014.

[8] McKay, P., Carriveau, R., and Ting, D. S.-K.: Wake impacts on downstream wind turbine performance and yaw alignment,Wind Energy, 16, 221–234, doi:10.1002/we.554, 2013.

---

## Author Comment (AC2) · 8 Jun 2018

**Authors' response to Referee #2:**

We would like to thank the referee for reviewing this manuscript, the constructive feedback and the valuable comments. At this stage, we respond to referee #2's comments and suggest changes for the final manuscript. The referee's original comments are printed in **bold** followed by the corresponding answers. Passages from the manuscript are printed in *italic writing*, in which proposed additions are indicated in blue and deleted parts in red.

Thank you very much for your efforts,

Jan Bartl on behalf of all authors
* * *
**Major comment (1)**

**Figure 8(c): I found it very surprising that, for large lateral offset values such as 0.16 and 0.33 (normalized with the rotor diameter), the yaw moment of the downwind turbine is higher when the first turbine is yawed. On the contrary, I expect to see a lower moment in this case as the wake deflection essentially alleviates partial wake conditions.**

This is a good comment and may indeed seem surprising in the first place. In order to judge the exact inflow conditions to the downstream turbine, we need to take a closer look into the wake flow of the upstream turbine at $x/D = 3$ (plots taken from Bartl et al., 2018).

The red and pink circles in Figure 1 (a) and (b) indicate the locations of an imaginary downstream turbine operated at a lateral offset of $z/D = +0.16$ and $z/D = +0.33$, respectively. In Figure 1 (a) it can be observed a downstream turbine is still exposed to an almost full wake impingement for an offset of $x/D = +0.16$ and $\gamma_{T1} = 0°$, and therefore experiences a rather small yaw moment if $M_y^* \approx 0.012$ (Figure 8(c) of the manuscript) in this situation. At $x/D = 3$, they wake has slightly expanded to an area, which is wider than the rotor swept area. Even for a lateral offset of $z/D = +0.33$, the major part of the downstream turbine's rotor swept area (pink circle, Figure 1 (a)) is impinged by the low velocity field of the wake, while only about the outer 3rd of the blade tips pass the high velocity freestream flow outside the wake.

For an upstream turbine yaw angle of $\gamma_{T1} = 30°$, as shown in Figure 1 (b), the wake flow is significantly deflected. However, at this rather small downstream distance, the wake is not entirely deflected away from a downstream turbine. For both lateral offset positions $z/D = +0.16$ and $z/D = +0.33$ of the downstream turbine, roughly half of the rotor swept area is impinged by the low velocity wake, while the other halve is impinged by high velocity freestream flow. Consequently, very high yaw moments of $M_y^* \approx 0.042$ are measured for both situations (Figure 8(c) of the manuscript). At an even higher lateral offset of $z/D = +0.50$, the yaw moments are observed to decrease. But still, the wake cannot be entirely deflected away for this large offset $z/D$ and small separation distance $z/D$.

[Figure]

Figure 1: Mean streamwise velocity $\overline{u}/u_{ref}$ in a cross-sectional cut at $x/D = 3$ through the wake flow behind a single turbine for **(a)** $\gamma_{T1} = 0°$ and **(b)** $\gamma_{T1} = 30°$. The red and pink circles indicate the locations of an imaginary downstream turbine operated at a lateral offset of $z/D = +0.16$ and $z/D = +0.33$, respectively. The plots are adapted from Bartl et al. (2018) and were measured behind the same model turbine under the same boundary conditions.
* * *
**Minor comment (1)**

**It would be useful to mention that the yaw moment can only be an indicator of unsteady loads due to inflow shear or yaw misalignment. The effect of large turbulent structures (especially those in atmospheric boundary-layer flows) on turbine loads cannot be shown by the sole consideration of yaw moment.**

This is indeed something that should be discussed in more depth. As already mentioned by reviewer #1, the connection between yaw moments and unsteady blade loads should be commented on in the introduction. We therefore suggest the following addition to the introduction in the manuscript:

p.3, l.19 f:
*For this purpose the parameters turbine separation distance $x/D$, lateral turbine offset $z/D$ and turbine yaw settings $\gamma_{T1}$ and $\gamma_{T2}$ are systematically varied in this wind tunnel experiment. Aside from power output and rotor thrust, the yaw moments acting on the individual rotors are measured. Yaw moments are a representation of the imbalance of the forces acting on a rotor blade during the course of one rotation. High values of yaw moments thus indicate increased unsteady blade loading at a frequency corresponding the rotational speed. Special focus is given to the concept of downstream turbine yawing (...).*

**Minor comment (2)**

**I agree with the other reviewer that the discussion part is relatively redundant, and it does not add new contribution to the paper.**

As mentioned in the answer to reviewer #1 already, we agree that the discussion mainly repeats previously presented results and only sparsely provides new information. We therefore will completely omit the Discussion section in the final version of the manuscript. References to external sources will be moved from the Discussion to the Results section. This concerns the following sections:

p.7, l.3 f:
*These asymmetries are slightly stronger for inflow A ($TI_A = 0.23\%$). Although it is not entirely clear where these stem from, the only reasonable source for an asymmetric load distribution in an uniform inflow is the rotor's interaction with the turbine tower. In the course of a revolution, the blades of a yawed turbine experience unsteady flow conditions, i.e. fluctuations in angle of attack and relative velocity. When superimposing an additional low-velocity zone, tower shadow or shear for example, the yaw-symmetry is disturbed. Asymmetric load distributions for turbines exposed to sheared inflow were recently reported by Damiani et al. (2017). They showed that vertical wind shear causes asymmetric distributions of angle of attack and relative flow velocity in the course of a blade revolution. They link these to rotor loads and conclude further consequences on wake characteristics and wind farm control strategies.*

p.10, l.14 f:
*Relative power gains of about 11% were measured at Inflow A, while only 8% were obtained for Inflow B at the same yaw angle of $\gamma_{T1} = -30°$. Asymmetries in the combined power output have been previously observed in a computational study Gebraad et al. (2016) and a similar experimental setup by Schottler et al. (2015). In a recent follow-up study, Schottler et al. (2017) attributed the asymmetry to a strong shear in the inflow to the two-turbine setup. As the inflow in the present study was measured to be spatially uniform, inflow shear is not a reason for the observed asymmetries.*

p.14, l.3 ff:
*In conclusion, is has been demonstrated that intentional upstream turbine yaw control is favorable in offset situations when considering both, the power output and yaw moments on a downstream turbine. Depending on the downstream turbine's streamwise and lateral position, the wake can be partly or even fully deflected away from its rotor swept area. This finding experimentally confirms results of a similar test case recently computed with a model-framework by van Dijk et al. (2017).*

p.14, l.18 f:
*Simultaneously, the yaw moment is measured to be around zero at this yaw angle. The potential of load reductions of a single turbine by yawing has been previously discussed by Kragh and Hansen (2014), in situations where the rotor was exposed to vertically sheared inflows. In the present test case, however, the partial wake impingement on the rotor represents a situation of a strongly horizontally sheared flow. Whether the shear in the incoming wind field is horizontal or vertical obviously makes a big difference, but mitigation of loads and maximization of power might be possible with yaw adjustments in both cases.*

p.14, l.20 f:

*The simultaneous power increase for the oppositely yawed downstream rotor is a positive side effect, although the exact reasons for the power increase are not entirely clear at this stage. A power increase by downstream turbine yawing has previously been reported in a full-scale data evaluation by McKay et al. (2013), who found an offset in the downstream turbine's yaw alignment for the purpose of optimized power output when operated in a partial wake of an upstream turbine. The downstream turbine yaw angle was observed to adjust itself opposed to the velocity gradient in the partial wake impinging the downstream rotor. These findings are in total agreement with the optimal downstream turbine yaw angle measured in our wind tunnel experiment.*

**Minor comment (3)**
**Please compare your wind tunnel blockage ratio with commonly acceptable values in the literature.**

This is a very good comment, which points to one of the weaknesses of the presented study. Commonly, a solid body should block less than 10% of the wind tunnel's cross sectional area. However, the blockage of a wind turbine rotor is dependent on the tip speed ratio. Dedicated studies investigating the influence of blockage on the performance of a wind turbine have been proposed by Sørensen et al. (2006) and Ryi et al. (2015). The proposed models are able to correct the power output of a single turbine. For an array of two aligned (and especially offset) turbines, no models have been developed yet to our knowledge. Recently, a dedicated computational study on the influence of the blockage ratio on the wake development for different inflow conditions was presented by Sarlak et al. (2016). In this study, a significant influence on the wake expansion was observed for a blockage ratio of 20%. In the present study, we intentionally do not use any blockage correction models, as we do not want to add another dimension of modeling uncertainty to our results. We are aware that our results do not represent a realistic, unblocked, full-scale wind turbine test case. They rather represent a model test case in defined boundary conditions, which can be used as a reference case for computational studies. In order elaborate more on this, we suggest to add the following lines to the manuscript:

p.4, l.5 ff:
*Moreover, about 12.8% of the wind tunnel's cross sectional area are blocked by the turbines' rotor swept area. The wind tunnel width measures about three times the turbine's rotor diameter, which leaves sufficient space for lateral wake deflection and offset positions for T2. However, a speed-up of the flow in free-stream areas around the rotors is observed due to blockage effects as described in detail in Bartl et al. (2018). The impact of the wind tunnel blockage on the wake expansion behind the same model turbine rotor has furthermore been investigated in a computational study by Sarlak et al. (2016). For high blockage ratios, correction models e.g. by Sørensen et al. (2006) or Ryi et al. (2015) for the power output are available. In this study, however, no correction models have been applied, in order not to add another dimension of modeling uncertainty to the results.*

**Minor comment (4)**
**Page 6, Lines 28 and 29: Please compare your results with those reported in the literature (e.g., Ozbay et al. 2012 and Bastankhah and Porté-Agel 2017).**

Thank you for this valuable comment. This is indeed still a widely discussed topic in research, and should be discussed in more detail. Four more external sources are referred to for comparison:

p.6, l.28 f:
*As discussed by Bartl et al. (2018), the decrease in power coefficient can be approximated $C_{P,\gamma_{T1}=0} \cdot cos^3(\gamma_{T1})$ when the turbine yaw angle is varied. The thrust coefficient's reduction through yawing is observed to match well with $C_{T,\gamma_{T1}=0} \cdot cos^2(\gamma_{T1})$. Despite the commonly assumed exponent of $\alpha = 3$ for the power coefficient $C_P(\gamma) = C_{P,\gamma=0} \cdot cos^\alpha$, Micallef and Sant (2016) refer to different values of $\alpha$ between 1.8 and 5 measured in different full-scale tests. The measured relations of our study, however, correspond well with previous measurements on the same rotor by Krogstad and Adaramola (2012) and another experimental study on a smaller rotor by Ozbay et al. (2012). Another recent experimental study on a very small rotor by Bastankhah and Porté-Agel (2017) confirmed the $\alpha = 3$ for the power coefficient, but found an slighly smaller exponent of $\beta = 1.5$ for the thrust coefficient.*

**Minor comment (5)**
**Page 4, Line 17: Please add space between ": : : still detectable." and "At".**

Thank you for the hint. The typing mistake is fixed for the final version of the manuscript.

p.4, l.16 ff:
*A velocity variation of $\pm 2.5\%$ is measured at $x/D = 0$ for Inflow B, as the footprint of the grid's single bars are still detectable. At $x/D = 3$, however, the grid-generated turbulent flow is seen to be uniform...*

**Minor comment (6)**
**Page 5, Line 15: It should be written as ": : : and 0.007 (0.9% of the absolute CT value), respectively".**

Thank you for pointing at this. This will be fixed in the final version of the manuscript.

p.5, l.14 f:
*The total uncertainties in power and thrust coefficient are 0.006 (2.5% of the absolute $C_P$-value)  0.007 (0.9% of the absolute $C_T$-value), respectively.*

**Minor comment (7)**
**Figure 3: I recommend using colors with more contrast.**

Thank you for this legitimate comment. We agree that the different shades of green and blue are not well distinguishable in the plot. However, the use of different symbols should make it possible to identify the curves corresponding to the different yaw angles.

**Minor comment (8)**
**Page 9, Line 9: Can it be shown using velocity measurements?**

This is a good comment, which has been pointed to by reviewer #1 as well. We suggest to add a some text explaining the effects of the wall blockage on the freestream velocity outside of the wake in more detail. Wake flow measurements showing this effect are presented in a previous publication on Wind Energy Science (companion paper) by Bartl et al. (2018).

p.9, l.7 ff:
*These high downstream power coefficients $C_{P,T2}$ can be explained by increased velocity levels of $u/u_{ref} = 1.10$ in the freestream outside of the wake as a result of wind tunnel blockage (Bartl et al., 2018). The downstream turbine power coefficient is, however, still referred to the undisturbed far upstream reference velocity $u_{ref}$. Although a considerable part of the downstream turbine rotor is impinged by T1's wake,  higher wind speeds outside of the wake lift the downstream turbine's power to these levels.*

**Minor comment (9)**
**Additional references:**
**Ozbay, A., Tian, W., Yang, Z. and Hu, H., 2012. Interference of wind turbines with different yaw angles of the upstream wind turbine. In 42nd AIAA Fluid Dynamics Conference and Exhibit (p. 2719).**
**Bastankhah, M. and Porté-Agel, F., 2017. Wind tunnel study of the wind turbine interaction with a boundary-layer flow: Upwind region, turbine performance, and wake region. Physics of Fluids, 29(6), p.065105.**

Thank you for alluding these two valuable references. They have been included to the manuscript in the discussion of the dependency of the power and thrust coeffient on the yaw angle (see Minor comment (4)).

**References**

[1] Bartl, J., Mühle, F., Schottler, J., Hölling, M., Peinke, J., Adaramola, M., and Sætran, L.: Wind tunnel experiments on wind turbine wakes in yaw: Influence of inflow turbulence and shear, Wind Energ. Sci., 3, 329–343, doi:10.5194/wes-3-329-2018, 2018.

[2] Sørensen, J.N., Shen, W.Z., Mikkelsen, R.: Wall correction model for wind tunnels with open test section, AIAA J., 44 (8), 1890–1894, 2006.

[3] Ryi, J., Rhee, W., Hwang, U.C., Choi, J.-S.: Blockage effect correction for a scaled wind turbine rotor by using wind tunnel test data, Renewable Energy, 79, 227–235, doi:10.1016/j.renene.2014.11.057, 2015.

[4] Sarlak, H., Nishino, T., Martinez-Tossas, L.A., Meneveau, C., and Sørensen, J.N.: Assessment of blockage effects on the wake characteristics and power of wind turbines, Renewable Energy 93, 340–352, doi: 10.1016/j.renene.2016.01.101, 2016.

[5] Micallef, D. and Sant, T.: A Review of Wind Turbine Yaw Aerodynamics, Chapter 2 in Wind Turbines - Design, Control and Applications, InTech, doi: 10.5772/63445, 2016.

[6] Krogstad, P.-AA. and Adaramola, M. S.: Performance and near wake measurements of a model horizontal axis wind turbine, Wind Energy, 15, 743–756, doi:10.1002/we.502, 2012.

[7] Ozbay, A., Tian, W., Yang, Z. and Hu, H.: Interference of wind turbines with different yaw angles of the upstream wind turbine, AIAA Fluid Dynamics Conference and Exhibit, AIAA 2012-2719, 2012.

[8] Bastankhah, M. and Porté-Agel, F.: Wind tunnel study of the wind turbine interaction with a boundary-layer flow: Upwind region, turbine performance, and wake region. Physics of Fluids, 29, 65105, 2017.

---

## Author Comment (AC3) · 8 Jun 2018

**Authors' response to Referee #3:**

We thank the referee for reviewing this manuscript and appreciate the constructive feedback and the improving comments. At this stage, we answer to referee #3's comments and propose changes for the final manuscript. The referee's original comments are printed in **bold** followed by the corresponding answers. Passages from the manuscript are printed in *italic writing*, in which proposed additions are indicated in blue and deleted parts in red.

Thank you very much for your efforts,

Jan Bartl on behalf of all authors

**Comment (1)**

Page 5, Line 14-15: The sentence should be rewritten as follows: "The total uncertainties in power and thrust coefficient are 0.006 (2.5% of the absolute 15 CP -value) and 0.007 (0.9% of the absolute CT-value), respectively."

Thank you for the hint. We will change the sentence in the final version of the manuscript.

**p.5, l.14 f:**

The total uncertainties in power and thrust coefficient are 0.006 (2.5% of the absolute  $C_P$ -value) respectively 0.007 (0.9% of the absolute  $C_T$ -value), respectively.

**Comment (2)**

Page 8, line 2-3: "The asymmetric wake deflection is considered to be the main reason for the asymmetric distribution of T2's yaw moments.". It is quite clear that yawing the upstream wind turbine in two different direction leads to different power gain on the downstream one. The authors trace back this behavior to not-well specified asymmetric wake deflection. It would be interesting, for the readers, if the authors could provide a deeper insight into this topic, considering that the authors (previously cited publication) already measured the wake shed by the upstream WT for two different yaw misalignment. Is the observed asymmetry due to asymmetric wake displacement or wake recovery?

Thank you for this very constructive comment. This is indeed one of the most important observations in this publication, and we agree that the underlying reasons for the asymmetry require a more detailed explanation. A previous publication by Bartl et al. (2018) discussed the asymmetries in wake displacement in detail, but we consider it to be important to revive the main reason for the asymmetric wake deflection here. For clarification, the following changes are suggested for the manuscript:

p.8, l.2 f:

The asymmetric wake deflection for positive and negative yaw angles is considered to be the main reason for the asymmetric distribution of T2's yaw moments. As discussed in an analysis of the wake flow behind a yawed turbine by Bartl et al. (2018), the overall wake displacement for positive and negative yaw angles was observed to be slightly asymmetric. The interaction of the rotor wake with the turbine tower was identified to be the main contributor for the asymmetric wake flow. This finding is supported by a previous study on the non-yawed wake by Pierella and Sætran (2017), in which they attributed a significant displacement of the wake center to the interaction with the turbine tower.

**Comment (3)**

Page 9, line 4-5: "Moreover, the downstream turbine's power output at low inflow turbulence (inflow A) is observed to be more asymmetric with respect to T1 than at high inflow turbulence (B)." This is quite surprising since one would expect that as more homogenous the flow is, as higher the symmetry of the phenomena is. It would be interesting if the authors could argue more about the reasons behind the observed data.

This is a very good comment, that also needs some more detailed explanation in the text. As the downstream turbine is operated in the partial wake of the upstream turbine, the inflow to the downstream turbine is no longer homogeneous. As shown in the analysis of the wake flow in Bartl et al. (2018), the deflection of the wake for positive and negative yaw angles is more asymmetric for an inflow of low turbulence (Inflow A). This can be qualitatively observed in the comparison of the mean wake flow at x/D = 6 presented in Figure 1 below. For a quantification of the shape and deflection of the mean wake flow for different inflow conditions, it is referred to Figure 7 and Figure 9 in Bartl et al. (2018). In order to make a clearer connection to the asymmetries in the incoming wake flow, the following modifications in the text are suggested:

**p.9, l.4 ff:**

Moreover, the downstream turbine's power output at low inflow turbulence (inflow A) is observed to be more asymmetric with respect to than at high inflow turbulence (inflow B). Especially for x/D = 6, the downstream turbine power  $C_{P,T2}$  is strongly asymmetric for inflow A. This observation corresponds well with the asymmetry in the mean streamwise wake flow measured for positive and negative yaw angles reported in Bartl et al. (2018). Therein, the wake flow behind a positively and negatively yawed turbine exposed to inflow A was observed to feature a higher degree of asymmetry than for the same turbine exposed to inflow B.

Figure 1: Mean streamwise velocity  $\overline{u}/u_{ref}$  in cross-sectional cuts at x/D = 6 through the wake flow behind a single turbine for  $\gamma_{T1} = 30^{\circ}$  and  $\gamma_{T1} = -30^{\circ}$  for inflow conditions A (upper row) and B (lower row). The plots are adapted from Bartl et al. (2018) and were measured behind the same model turbine under the same boundary conditions.

**Comment (4)**

Page 10. In a previous sentence, the authors reported that quite substantial wake blockage was observed, leading to an increase of 10% of the speed outside the wake of the upstream model. How much is the blockage affecting the results presented in Figure 5? Moreover, the rotor speed of the upstream model was kept constant even for a very high yaw misalignment, which implies that the upstream model is operating at sub-optimal conditions. Indeed, when yawing a wind turbine it would have been better to keep constant the effective TSR, i.e. the TSR computed by using the component of the wind speed orthogonal to the rotor disk. How much power is lost, on the upstream model, due to the fact the model itself is operating, while yawed, at sub-optimal conditions? How this affects the results presented in figure 5?

Answer to first part of the question (how much blockage affects results): This is a very good comment, which points to one of the main weaknesses of the present study. In general, it is very difficult to quantify, how much the blockage of the wind tunnel walls affects the combined power results. For this study, we have not tried to use any kind of blockage correction models on our results.

It would be possible to correct the power and thrust of a single turbine operated in a wind tunnel. Different models have been proposed by, amongst others, Sørensen et al. (2006) and Ryi et al. (2015). However, wind tunnel blockage possibly also affects the deflection and expansion of the wake flow, which is more difficult to correct. A dedicated study on the effects of blockage on the wake development was presented by Sarlak et al. (2016). In this study, a significant influence on the wake expansion was observed for a blockage ratio of 20%. The third and most difficult component of an assessment of the effects of blockage on the performance of a turbine array would be the performance of the downstream turbine operated in a (partial) wake of an upstream turbine. To our knowledge, there are currently no correction models available for this rather complex case. A comparative computational study of our setup in a domain, which includes and also omits the wind tunnel boundaries could be performed to shed light on this problem.

We are aware that our results do not represent a realistic, unblocked, full-scale wind turbine test case. They rather represent a model test case in defined boundary conditions, which can be used as a reference case for computational studies. In order elaborate more on this, we suggest to add the following lines to the manuscript:

**p.4, l.5 ff:**

Moreover, about 12.8% of the wind tunnel's cross sectional area are blocked by the turbines' rotor swept area. The wind tunnel width measures about three times the turbine's rotor diameter, which leaves sufficient space for lateral wake deflection and offset positions for T2. However, a speed-up of the flow in free-stream areas around the rotors is observed due to blockage effects as described in detail in Bartl et al. (2018). The impact of the wind tunnel blockage on the wake expansion behind the same model turbine rotor has furthermore been investigated in a computational study by Sarlak et al. (2016). For high blockage ratios, correction models e.g. by Sørensen et al. (2006) or Ryi et al. (2015) for the power output are available. In this study, however, no correction models have been applied, in order not to add another dimension of modeling uncertainty to the results.

Answer to second part of the question (how much additional upstream tur**bine TSR-control would affect results**): Also this second part of the question is a very good comment. A similar comment was given by reviewer #1. We have measured the operating characteristics of the upstream turbine in dependence of the yaw angle and tip speed ratio. For  $\gamma_{T1} = 0^{\circ}$  and  $\pm 30^{\circ}$  the operating characteristics for all inflow conditions are shown in the previous publication (Bartl et al., 2018), which already is referred to in the text. The complete characteristics for  $\gamma_{T1} = 0^{\circ}$  to  $+40^{\circ}$  (Inflow B) are shown here in Figure 2 for positive yaw angles only (note that negative yaw angles have an insignificantly higher magnitude, but very similar TSR-dependency). It can observed that the maximum power coefficient is measured at  $\lambda = 6.0$  for yaw angles between  $0^{\circ}$  and  $30^{\circ}$ . For the highest yaw angle of  $40^{\circ}$ , however, the optimum tip speed ratio is found at  $\lambda = 5.5$ , which makes sense according to the reasoning given by the reviewer. At this extreme yaw angle, a slightly higher combined power output could indeed have been achieved, if the upstream turbine would have been operated at  $\lambda = 5.5$ . However, a constant upstream turbine tip speed ratio of  $\lambda = 6.0$  seems to be optimum for the most interesting region between  $0^{\circ}$  and  $30^{\circ}$ . In conclusion, we think that only the results for the extreme yaw angles of  $\pm 40^{\circ}$  could slightly be affected by a non-optimum TSR control of the upstream turbine (ref. Figure 5 of the manuscript). For all other yaw angles, the upstream turbine was operated very close to its optimum.

Nevertheless, we suggest to add some additional lines of text to the manuscript discussing the TSR-dependency.

---

## Author Response (AR1)

[revised manuscript text omitted]
$        | 0.23%                | [-40°,, +40°]           | 3 & 6                                 | 0                           | $0^{\circ}$ $0^{\circ}$ |
| 1 (b) Aligned turbines                                                              | $\gamma_{T1}$ & $x/D$        | 10.0%                | [-40°,, +40°]           | 3 & 6                                 | 0                           |                         |
| <li>2 (a) Offset turbines</li><li>2 (b) Offset turbines</li>               | $\Delta z/D$                 | 10.0%                | 0°                      | 3                                     | [-0.5,+0.5]                 | $0^{\circ}$             |
|                                                                                     | $\Delta z/D$                 | 10.0%                | +30°                    | 3                                     | [-0.5,+0.5]                 | $0^{\circ}$             |
| <li>3 (a) Downstream turbine yaw</li><li>3 (b) Downstream turbine yaw</li> | $\Delta z/D$ & $\gamma_{T2}$ | 10.0%                | 0°                      | 3                                     | [-0.5,+0.5]                 | [-30°,,+30°]            |
|                                                                                     | $\Delta z/D$ & $\gamma_{T2}$ | 10.0%                | +30°                    | 3                                     | [-0.5,+0.5]                 | [-30°,,+30°]            |

[revised manuscript text omitted]